# The intracellular bacterium *Orientia tsutsugamushi* uses the autotransporter ScaC to activate BICD adaptors for dynein-based motility

Giulia Manigrasso [1], Kittirat Saharat[2], Panjaporn Chaichana [2], Chitrasak Kullapanich[3], Sharanjeet Atwal[4], Jerome Boulanger[1], Tomos E. Morgan[1], Holger Kramer[1,8], Jeanne Salje [2,3,5,6,7] ✉ & Andrew P. Carter [1,7] ✉

The intracellular bacterium *Orientia tsutsugamushi* relies on the microtubule cytoskeleton and the motor protein dynein to traffic to the perinuclear region within infected cells. However, it remains unclear how the bacterium is coupled to the dynein machinery and how transport is regulated. Here, we discover that *O. tsutsugamushi* uses its autotransporter protein ScaC to recruit the dynein adaptors BICD1 and BICD2 for movement to the perinucleus. We show that ScaC is sufficient to engage dynein-based motility in the absence of other bacterial proteins and that BICD1 and BICD2 are required for efficient movement of *O. tsutsugamushi* during infection. Using TIRF single-molecule assays, we demonstrate that ScaC induces BICD2 to adopt an open conformation which activates the assembly of dynein-dynactin complexes. Our results reveal a role for BICD adaptors during bacterial infection and provide mechanistic insights into the life cycle of an important human pathogen.

Many intracellular bacteria hijack the host cytoskeleton to acquire motility for their infection cycle. Whereas the most common strategy involves manipulation of actin assembly[1], a small group of bacteria, including *Chlamydia trachomatis* and *Orientia tsutsugamushi*, have developed the ability to exploit microtubules[2–4]. Although *Chlamydia* remains encased within an inclusion membrane derived from the endocytic compartment and leverages the host vesicular transport system, *O. tsutsugamushi* resides free in the cytosol and therefore needs to establish direct connections with microtubule motors. Because of this *O. tsutsugamushi* is a fascinating model to study the interplay between intracellular pathogens and the microtubule cytoskeleton.

*O. tsutsugamushi* is an obligate intracellular bacterium which is transmitted to mammalian hosts through the bite of mite larvae[5,6].

Infection in humans causes the potentially life-threatening disease scrub typhus, whose symptoms include fever, skin rash, myalgia and, in severe cases, multiple organ failure[6–8]. *O. tsutsugamushi* colonizes a range of host cell types, including endothelial cells and leukocytes of the innate immune system[9,10]. Entry into host cells occurs via clathrin-mediated endocytosis and macrophagocytosis and is followed by rapid disruption of the endosomal membrane to release the bacteria into the cytosol[11,12]. Cytosolic *O. tsutsugamushi* are transported along microtubules to the perinuclear region, where bacterial replication takes place[2]. Exit from infected host cells involves a budding-like mechanism, akin to the exit strategy employed by enveloped viruses, which releases membrane-enclosed bacteria into the extracellular space[13,14].

[1]MRC Laboratory of Molecular Biology, Cambridge CB2 0QH, UK. [2]Cambridge Institute for Medical Research, University of Cambridge, CB2 0XY Cambridge, UK. [3]Mahidol-Oxford Tropical Medicine Research Unit, Faculty of Tropical Medicine, Mahidol University, Bangkok 10400, Thailand. [4]Lonza, 8830 Biggs Ford Road, Walkersville, MD 21793, USA. [5]Department of Pathology, University of Cambridge, Cambridge CB2 1QP, UK. [6]Department of Biochemistry, University of Cambridge, Cambridge CB2 1GA, UK. [7]These authors contributed equally: Jeanne Salje, Andrew P. Carter. [8]Deceased: Holger Kramer. ✉e-mail: jss53@cam.ac.uk; cartera@mrc-lmb.cam.ac.uk

Transport of *O. tsutsugamushi* to the perinucleus requires the motor protein dynein and its cofactor dynactin[2], which together drive retrograde movement of many cellular cargoes. Dynein transport typically requires an activating adaptor, a coiled-coil component that activates dynein/dynactin and simultaneously links them to their cargos[15,16]. Over 15 such adaptors have been identified, each specific to a subset of cargos[17,18]. Although dynein's role in *O. tsutsugamushi* transport is established, it remains unclear which bacterial effectors are involved and whether association with dynein is direct or mediated by host factors like activating adaptors.

Here, we show that *O. tsutsugamushi* uses the autotransporter surface protein ScaC, which sits on the bacterial outer membrane[19–21], to recruit the dynein activating adaptors BICD1 and BICD2. Ectopically expressed ScaC engages dynein in the absence of other bacterial proteins, suggesting it is sufficient for the initiation of motility. We use size exclusion chromatography and single-molecule motility assays to demonstrate that ScaC promotes an open conformation of BICD2, which in turn facilitates the interaction with dynein and dynactin. Finally, we show that BICD1 and BICD2 are required for efficient perinuclear migration of *O. tsutsugamushi*. Overall, our study uncovers a mechanism for bacterial motility along microtubules.

## Results

### The bacterial protein ScaC binds the activating adaptors BICD1 and BICD2

Since *O. tsutsugamushi* is not surrounded by host membranes, we reasoned that it must have developed ways to directly recruit dynein to its surface. Bacterial proteins that sit on the outer membrane are likely candidates for this process as they are easily accessible to host factors in the cytosol. Among these, members of the surface cell antigen (Sca) family of autotransporters[22] are especially interesting as they possess a large extracellular domain capable of interacting with the host cell machinery and have been implicated in actin-based motility in *Rickettsia* species[23,24]. Sca proteins comprise a conserved C-terminal translocator domain and a variable N-terminal passenger domain[19]. The translocator domain forms a pore in the bacterial outer membrane which allows the passenger domain to pass through for display on the cell surface[25]. To determine whether Sca proteins are involved in the interaction with dynein, we set out to investigate the host interactome of four Sca family members that are known to be ubiquitously conserved across *O. tsutsugamushi* strains: ScaA, ScaC, ScaD and ScaE[19] (Fig. 1A). To this end, we expressed their passenger domains fused to a N-terminal GFP tag in Flp-In HeLa cells and performed pull-downs followed by protein identification by mass spectrometry. None of the hits for ScaA, ScaD and ScaE had known links to dynein (Supplementary Fig. 1A, Supplementary Data 1–3). On the other hand, the passenger domain of ScaC (residues 33–223, hereafter referred to as ScaC) co-precipitated BICD1 and BICD2, which are paralogs and both members of the Bicaudal D (BICD) family of activating adaptors (Fig. 1B, Supplementary Data 4)[15,16,26]. ScaC also pulled down BICD1 and BICD2 from the lysates of monocytic (THP-1) and dendritic (MutuDC) cell lines, showing that this interaction is not restricted to HeLa models but also occurs in cell types targeted by *O. tsutsugamushi* (Supplementary Fig. 1B, C).

To confirm that ScaC can engage the dynein machinery to drive motility, we performed a mitochondrial relocation assay in HeLa cells. In this approach, a protein of interest is targeted to the mitochondrial outer membrane using a mitochondrial targeting sequence (MTS) and the subcellular localization of the organelles is monitored by fluorescence microscopy (Fig. 1C)[27,28]. An accumulation of mitochondria at the perinuclear region provides evidence that the candidate protein is a dynein effector[27,29,30]. We generated Flp-In T-REX HeLa cell lines expressing GFP[MTS] or GFP-ScaC[MTS] fusion proteins (Supplementary Fig. 2A). As a positive control, we used a GFP-BICD2N[MTS] line expressing

the constitutively active N-terminal fragment of BICD2 (residues 1–400, BICD2N) which elicits dynein-based motility and has been used extensively in mitochondrial relocation assays[27,28,30]. GFP[MTS] expression did not alter the distribution of mitochondria, which remained scattered in the cytoplasm as observed in wild type cells (Fig. 1D, E, Supplementary Fig. 2B, C). Conversely, GFP-ScaC[MTS] caused dramatic perinuclear clustering of the organelles, which closely resembled the effect of GFP-BICD2N[MTS] (Fig. 1D, E). Importantly, mitochondrial clustering was not caused by an indirect stress response to GFP-ScaC overexpression, as cells overexpressing ScaC without the MTS tag exhibited wild type growth rates and maintained normally dispersed mitochondria (Supplementary Fig. 2D, E). Furthermore, treatment with the microtubule depolymerizing agent nocodazole significantly reduced the perinuclear accumulation of mitochondria observed in GFP-ScaC[MTS] cells, confirming that ScaC-induced clustering relies on intact microtubules (Supplementary Fig. 2F, G).

Our pull-down experiments indicated an interaction between ScaC and the activating adaptors BICD1 and BICD2 (Fig. 1B). Accordingly, immunofluorescence analysis of GFP[MTS] and GFP-ScaC[MTS] cells showed that BICD1 and BICD2 are both recruited to mitochondrial clusters in a ScaC-dependent manner (Fig. 1F, G, Supplementary Fig. 2H). To determine if ScaC requires BICD1 and BICD2 for mitochondrial relocation, we performed knockdowns of both adaptors in GFP-ScaC[MTS] cells (Supplementary Fig. 2I). Depletion of BICD1 alone did not affect the ScaC-dependent clustering of mitochondria. However, knockdown of BICD2 strongly inhibited mitochondrial accumulation at the perinuclear region, and a similar phenotype was observed when a double BICD1/2 knockdown was performed (Fig. 1H, I). Notably, the effect of BICD2 knockdown on mitochondrial distribution was specific to GFP-ScaC[MTS] cells and was not observed in control GFP[MTS] lines (Supplementary Fig. 2J, K). Based on these results, we conclude that BICD2 acts as the predominant adaptor for ScaC in our mitochondrial relocation assay. The gene expression levels of *BICD2*, measured by qPCR, are ~57-fold higher than those of *BICD1* in the cell model used in these experiments (Supplementary Fig. 2L), which suggests that BICD2 is highly abundant and therefore able to compensate for BICD1 depletion. To test whether BICD1 and BICD2 are equally capable of mediating ScaC transport, we overexpressed an mCherry-BICD1 construct in BICD2-depleted HeLa GFP-ScaC[MTS] cells (Supplementary Fig. 2M, N). This effectively restored the perinuclear accumulation of mitochondria, demonstrating that BICD1 and BICD2 are functionally redundant and that their roles in ScaC transport likely depend on their relative abundance, which varies across cell types.

### ScaC interacts with the CC3 domain of BICD2

To determine whether ScaC recruits BICD2 directly, we purified both proteins and tested their ability to form a complex. Circular dichroism spectroscopy showed that the ScaC passenger domain adopts a classical α-helical conformation (Supplementary Fig. 3A). Similar to BICD2[31], ScaC behaves as a dimer in solution (Supplementary Fig. 3B, C), consistent with its predicted coiled-coil structure (Supplementary Fig. 3D). Size exclusion chromatography (SEC) showed that ScaC and BICD2 form a stoichiometric complex, indicating that they do not require additional factors to interact (Fig. 2A). Size-exclusion chromatography with multi-angle light scattering (SEC-MALS) analysis estimated a complex mass of 244 kDa which is consistent with a ScaC-BICD2 complex of 2:2 stoichiometry (Fig. 2B). Notably, BICD2 did not interact with the passenger domain of ScaE over SEC, highlighting the specificity of the ScaC-BICD2 interaction (Supplementary Fig. 3E).

To identify the ScaC-binding region on BICD2, we designed four BICD2 truncations and tested their ability to interact with ScaC (Fig. 2C, Supplementary Fig. 3F). ScaC bound to the C-terminus of BICD2 (BICD2[342–800]) but not to the N-terminus (BICD2[1–400]).

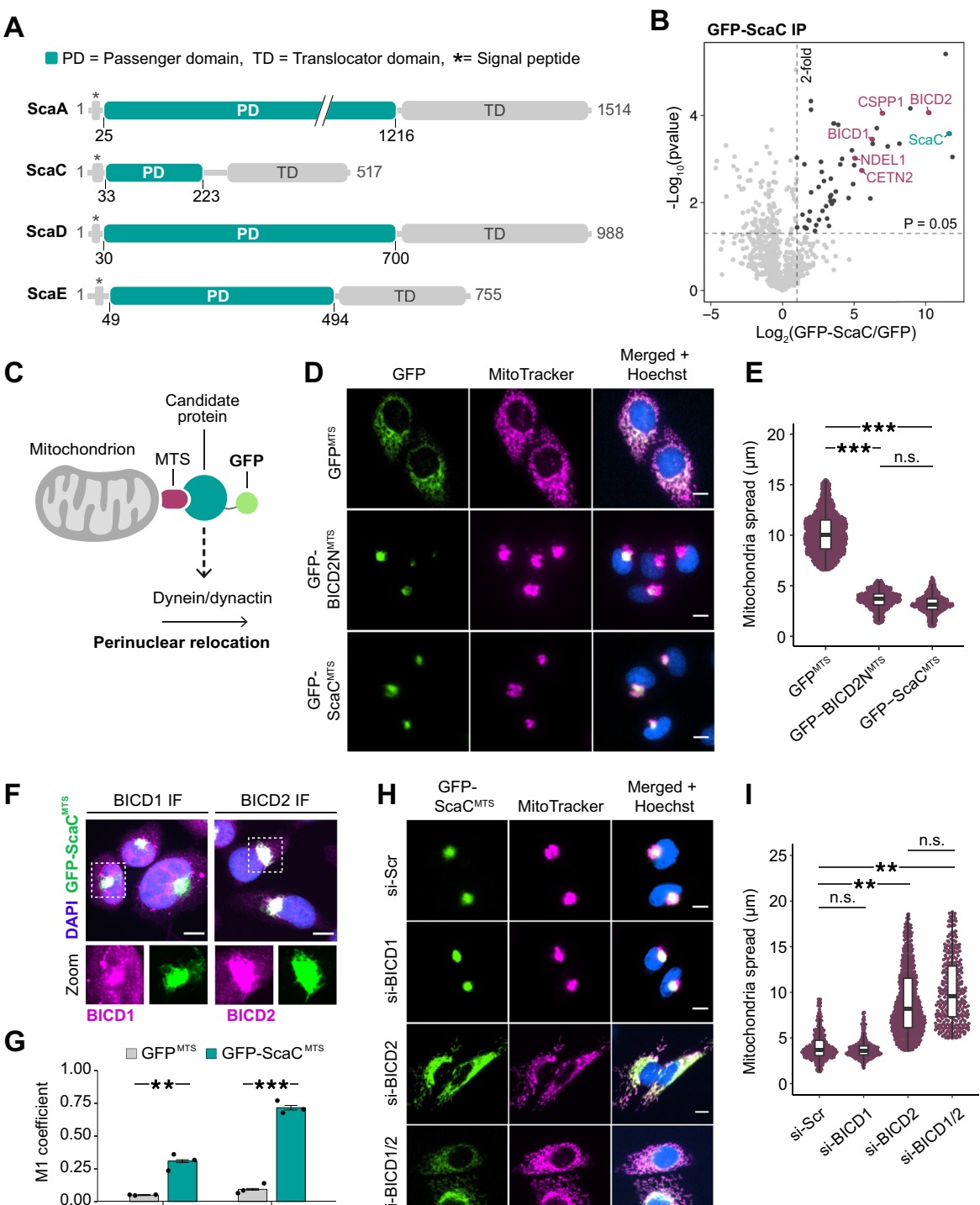

**Fig. 1 | The bacterial surface protein ScaC binds the activating adaptors BICD1 and BICD2. A** Domain architecture of conserved Sca proteins. The passenger domains are highlighted in green. **B** Volcano plot summarising the results of GFP-ScaC co-IP/MS in HeLa Flp-In cells. The significant enrichment levels were calculated from $n = 3$ independent replicates using a two-sided $t$-test. Peptide identifications were filtered for high confidence using the Percolator node (FDR < 1%, rank 1 peptides). GFP-ScaC is highlighted in green. Proteins with known links to microtubules are indicated in magenta. **C** Visual schematic of the mitochondrial relocation assay. **D** Representative images of mitochondrial relocation assay with GFP$^{MTS}$, GFP-BICD2N$^{MTS}$ and GFP-ScaC$^{MTS}$ cell lines. Scale bars indicate 10 µm. **E** Quantification of mitochondrial spread for GFP$^{MTS}$, GFP-BICD2N$^{MTS}$ and GFP-ScaC$^{MTS}$ samples. The beeswarm plot includes all values between 5th and 95th percentiles. The box plot centre indicates the median, the boxes show the 25th/75th percentile, and whiskers extend to the largest or smallest values within $1.5 \times$ IQR from the hinges. $P$-values were calculated from $n = 3$

independent replicates using two-tailed ANOVA with a Tukey HSD pot-hoc test. \*\*\*$p < 0.001$. GTP$^{MTS}$:GFP-BICD2N$^{MTS}$ $p = 6.2 \times 10^{-5}$, GFP$^{MTS}$:GFP-ScaC$^{MTS}$ $p = 5.1 \times 10^{-5}$ **F** Representative IF images showing colocalization of GFP-ScaC$^{MTS}$ with BICD1 and BICD2 respectively. Scale bars indicate 5 µm. **G** Barplot showing colocalization of BICD1 and BICD2 with GFP$^{MTS}$ (grey) or GFP-ScaC$^{MTS}$ (green). Colocalization was measured using the Manders coefficient (M1). $P$-values were calculated from $n = 3$ independent replicates using a two-tailed $T$-test. Error bars indicate mean ± standard error. \*\*$p < 0.01$, \*\*\*$p < 0.001$. BICD1 $p = 0.002$, BICD2 $p = 6.3 \times 10^{-5}$ (**H**) Representative images of GFP-ScaC$^{MTS}$ cells treated with siRNAs against BICD1 and/or BICD2. A scrambled siRNA (si-Scr) was used as a negative control. Scale bars indicate 10 µm. **I** Quantification of mitochondrial spread in siRNA-treated GFP-ScaC$^{MTS}$ cells. The beeswarm and box plots are as described in (**E**). $P$-values were calculated from $n = 3$ independent replicates using two-tailed ANOVA with a Tukey HSD pot-hoc test. si-Scr:siBICD2 $p = 0.005$, si-Scr:si-BICD1/2 $p = 0.002$. Source data are provided as a Source Data file.

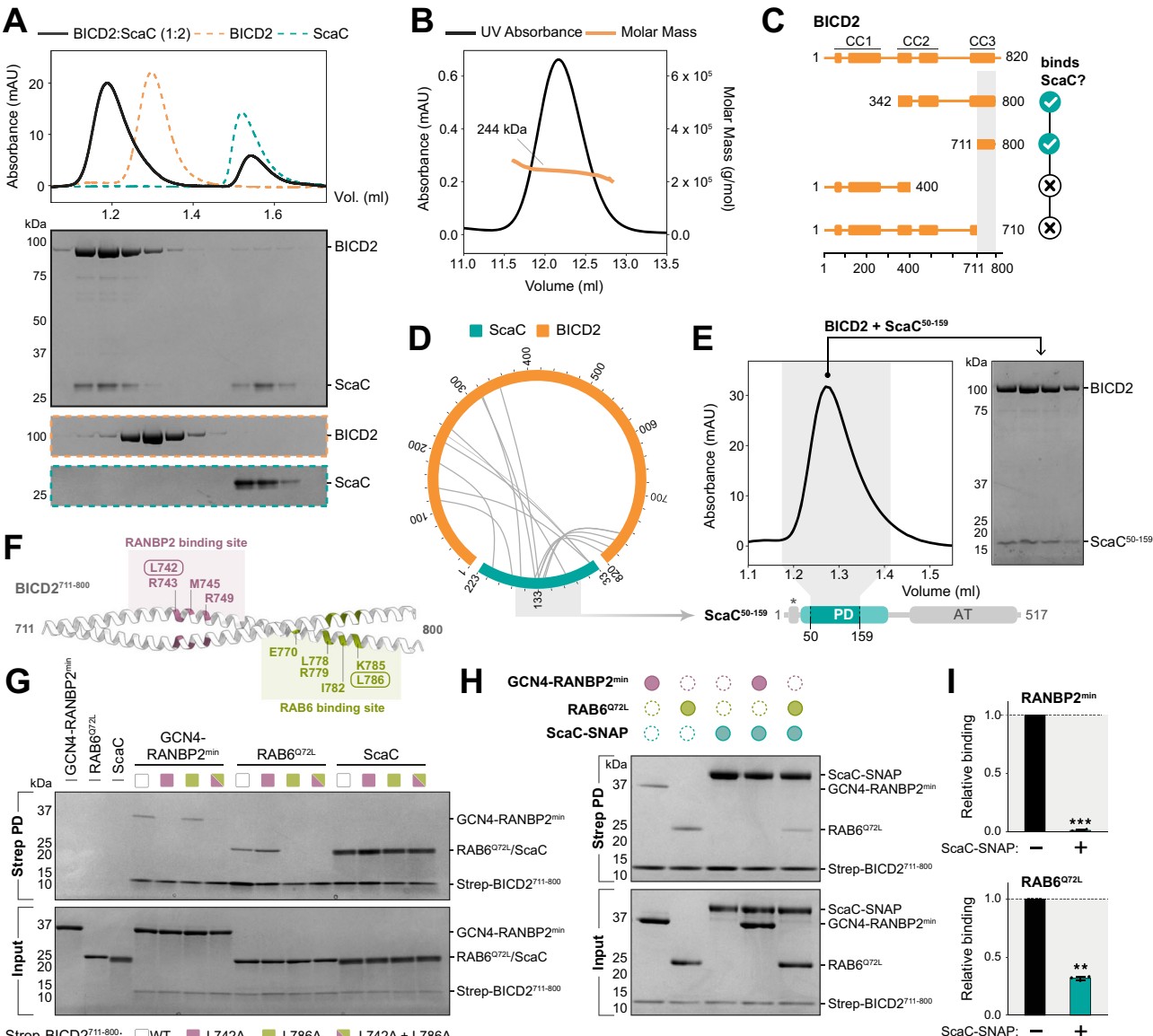

**Fig. 2 | ScaC binds the C-terminus of BICD2. A** Chromatogram for the BICD2-ScaC complex (black). Purified BICD2 and ScaC were mixed in 1:2 molar ratio. Individual traces for BICD2 (orange) and ScaC (green) are shown. SDS-PAGE gels show fractions underlying the relevant protein peaks. This SEC experiment was repeated three times (**B**) SEC-MALS for the BICD2-ScaC complex. The molar mass of the complex is indicated. **C** Schematic diagram showing mapping of the ScaC-binding domain (highlighted in grey) using BICD2 truncations. **D** Crosslinking/MS of purified BICD2-ScaC complexes using ECD. Heteromeric crosslinks are shown. **E** Chromatogram of the BICD2-ScaC[59-159] complex. The SDS-PAGE gel shows fractions underlying the complex peak. This SEC experiment was repeated three times (**F**) AlphaFold prediction of BICD2[711-800]. The residues involved in binding to

RANBP2 and RAB6[35] are highlighted in pink and green respectively. **G** SDS-PAGE gels showing in vitro pull-downs of different Strep-BICD2[711-800] mutants in the presence of 3.5-fold excess GCN4-RANBP2[min], RAB6[Q72L] or ScaC. This pulldown experiment was repeated three times (**H**) Competition pull-down assay using Strep-BICD2[711-800] in the presence of different combinations of GCN4-RANBP2[min], RAB6[Q72L] and ScaC-SNAP. **I** Quantification of GCN4-RANBP2[min] (top) and RAB6[Q72L] (bottom) binding to Strep-BICD2[711-800] in the absence or presence of ScaC-SNAP. Error bars indicate mean ± standard deviation. P-values were calculated from $n = 3$ independent replicates using a two-tailed T-test. **p < 0.01. RANBP2 $p = 8.8 \times 10^{-5}$. RAB6 $p = 0.0025$. Source data are provided as a Source Data file.

Furthermore, we found that the third C-terminal coiled-coil domain (CC3) of BICD2 (BICD2[711-800]) was both necessary and sufficient for the interaction with ScaC (Fig. 2C, Supplementary Fig. 3F). These findings were validated by crosslinking mass spectrometry of purified ScaC and BICD2 with the zero-length crosslinker EDC, which showed an enrichment of crosslinks around the CC3 of BICD2 (Fig. 2D, Supplementary Data 5). The CC3 crosslinks converged on the central region of the ScaC passenger domain, (residues 119–136), suggesting it contains the binding site for BICD2. In line with this observation, a truncated fragment of ScaC spanning residues 50–159

retained the ability to bind BICD2 (Fig. 2E). ScaC was also able to bind the CC3 of BICD1, which shares 94% homology to the CC3 of BICD2, suggesting that a shared mechanism underlies both interactions (Supplementary Fig. 3G).

**ScaC has evolved a distinct interface for BICD2 binding**
The host proteins RANBP2 and RAB6 also interact with the CC3 domain of BICD2 to facilitate the transport of their associated cargoes (Supplementary Fig. 4A)[31–33]. RANBP2 is a nucleoporin which recruits dynein to the nuclear envelope to drive separation of the

nucleus from the centrosomes during G2 phase[32], whereas RAB6 is a small GTPase protein which orchestrates the retrograde movement of Golgi-derived and secretory vesicles[33,34]. Although these two effectors employ similar mechanisms for mediating transport, they bind non-overlapping sites on BICD2 (Fig. 2F, Supplementary Fig. 4B)[35]. We set out to determine whether ScaC binds BICD2 via one of these sites or uses a different binding interface. Initially, we observed that ScaC does not bind BICDR1, another member of the Bicaudal D family which contains a conserved Rab6 binding site (Supplementary Fig. 4C, D)[36]. This suggests ScaC does not contact exactly the same site on BICD2 as RAB6. To follow up on this observation, we introduced mutations at two residues in the CC3 domain of BICD2, L742 and L786, which are essential for the interaction with RANBP2 and RAB6, respectively (Fig. 2F)[31,35]. To validate our mutants, we purified a RANBP2 fragment (residues 2148–2240 or RANBP2[min]), which contains the minimal BICD2 binding domain[37,38], and a GTP-locked RAB6 mutant (RAB6[Q72L]). As anticipated, the L742A mutation resulted in loss of RANBP2 binding in an in vitro pulldown assay, while the L786A mutation prevented RAB6[Q72L] binding. A double L742A/L786A mutant effectively abolished the interaction with both proteins (Fig. 2G). In contrast, we found that ScaC retained the ability to bind all mutant versions of BICD2 (Fig. 2G) suggesting it interacts with the adaptor in a different way from either RAB6 or RANBP2. Finally, we performed a competition pull-down assay to determine whether the ScaC binding site overlaps with those of RANBP2 or RAB6. In the presence of ScaC, RAB6 binding to BICD2 was decreased by 60% while RANBP2 binding was fully abrogated (Fig. 2H, I). This hints that ScaC occupies a region on BICD2 which broadly overlaps with the RANBP2 binding region and at least partially extends to the RAB6 interaction site.

## ScaC activates BICD2 for dynein motility

In the absence of a suitable cargo, BICD2 is folded over and auto-inhibited by an interaction of its C-terminal domain with its N-terminus, which effectively blocks the recruitment of dynein and dynactin[26,39]. To investigate whether ScaC binding to BICD2 is sufficient to relieve autoinhibition we used a previously reported assay in which the interaction between separate N- and C-terminal BICD2 constructs is assayed by SEC[26,35]. In the absence of ScaC, the BICD2[1–400] (BICD2N) and BICD2[342–800] (BICD2C) constructs interact and eluted together (Fig. 3A). Addition of ScaC resulted in the formation of a complex between ScaC and BICD2C, whereas BICD2N eluted separately in fractions corresponding to the protein alone (Fig. 3A). This suggests ScaC displaces the N-terminus of BICD2 to promote an active open conformation. To confirm this process is sufficient to stimulate dynein motility, we used single-molecule fluorescence microscopy to monitor the movement of TMR-labelled dynein along microtubules (Fig. 3B, C). In the presence of dynactin and the constitutively active cargo adaptor BICD2N (DDB[N]), dynein displayed robust directional movement. Conversely, full-length BICD2 produced fewer processive complexes (DDB), consistent with previous reports[40,41]. Adding ScaC to full-length BICD2 (DDBS) increased the number of processive events by 3-fold (for an average of $1.22 \pm 0.12$ vs. $0.44 \pm 0.12$ processive events/ µm MT/min). Importantly, the increase in dynein motility caused by ScaC was BICD2-dependent, since DDS only exhibited diffusive movement along microtubules (Fig. 3B, C). We did not observe any significant differences in velocity or run length between DDBS and DDB conditions, with average speeds ranging between 0.9 and 1.3 µm s[−1] and run lengths between 6.2 and 8.7 µm (Supplementary Fig. 5A, B). To verify that ScaC co-migrates with the active complex following BICD2 activation, we performed dual colour motility assays with ScaC-

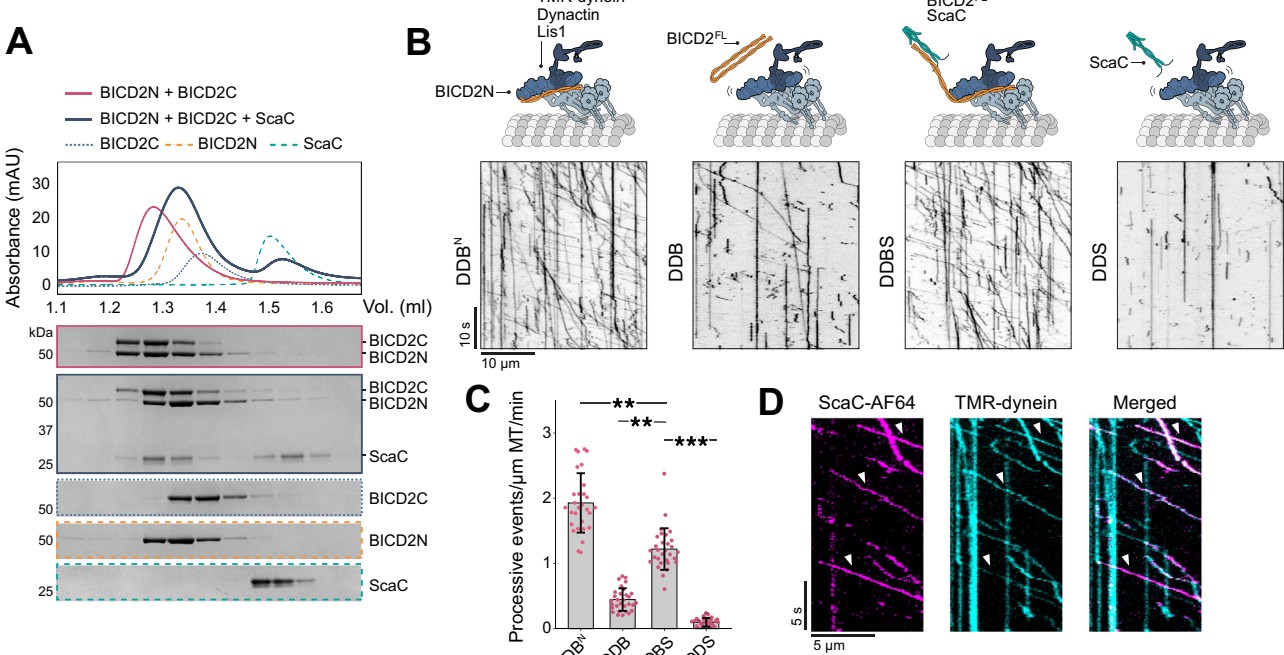

**Fig. 3 | ScaC activates BICD2 for dynein-based motility. A** SEC chromatograms showing that BICD2N and BICD2C form a complex (purple trace and gel) that is disrupted upon BICD2C binding to ScaC (blue trace and gel). Individual traces and gels (dotted lines) for BICD2N, BICD2C and ScaC show where these proteins elute on their own. SDS-PAGE gels show fractions underlying the relevant protein peaks. This SEC experiment was repeated three times (**B**) Representative kymographs of TMR-dynein mixed with dynactin and Lis1 in the presence of either BICD2N, full-length BICD2, full-length BICD2 and ScaC or ScaC only. **C** Barplot showing the number of processive events per microtubule µm per minute for different complex combinations (as in **B**). *P*-values were calculated from $n = 3$ independent replicates using two-tailed ANOVA with a Tukey HSD pot-hoc test. Error bars indicate mean ± standard deviation. ***p* < 0.01, ****p* < 0.001. DDBN:DDBS *p* = 0.009, DDB:DDBS *p* = 0.005, DDBS:DDS *p* = 0.0005 (**D**) Representative kymographs showing colocalization of Scac-AF674 (magenta) and TMR-dynein (cyan). Arrows indicate colocalization events. Scale bars are shown. Source data are provided as a Source Data file.

AF647 and TMR-dynein. We found that 74.8 ± 3.2% of moving dyneins colocalized with ScaC in DDBS complexes, thus demonstrating that ScaC is co-transported with the motile machinery (Fig. 3D). Similarly, colocalization was observed for ScaC-AF647 and BICD2-AF488 (Supplementary Fig. 5C).

## BICD1 and BICD2 are required for perinuclear translocation of *O. tsutsugamushi*

Having established that the interaction between ScaC and BICD2 is sufficient to elicit microtubule-based transport in cells and in vitro, we asked whether it is required for *O. tsutsugamushi* motility during infection. At 4 h post infection (hpi) we were able to detect occasional long distance (-2–20 μm), processive movements of individual bacteria with an average speed of 0.09 μm/ +/− 0.005 μm/s (Supplementary Movie 1, Supplementary Fig. 6A–C). However, the frequency of these events was low and therefore we opted to look instead at the accumulation of bacteria next to the nucleus 2 days post infection (dpi) using an antibody against the abundant *O. tsutsugamushi* surface protein TSA56. First, we confirmed a role for microtubules in *O. tsutsugamushi* movement by treating infected HeLa ATCC CCL-2 cells with nocodazole, which led to a loss of perinuclear accumulation of the bacteria (Fig. 4A, B). Next, we tested the importance of ScaC for *O. tsutsugamushi* localization by generating a cell line in which a GFP-ScaC construct was exogenously overexpressed. This would compete with native ScaC on the bacteria and phenocopy a ScaC deletion which is not technically possible since *O. tsutsugamushi* is genetically intractable. Although ScaC-GFP overexpression did not affect bacterial replication rates under these growth conditions (Supplementary Fig. 6D), it resulted in a loss of *O. tsutsugamushi* perinuclear localisation equivalent to that of nocodazole treatment (Fig. 4C, D). This suggests ScaC is required for *O. tsutsugamushi* motility.

We next determined whether BICD1/2 adaptors are used by *O. tsutsugamushi* during infection. Although the mRNA levels of BICD2 in the HeLa ATCC CCL-2 cells used in these experiments are around 30 times those of BICD1 (Supplementary Fig. 6E), our data presented above suggests that both adaptors can bind ScaC. Therefore, we proceeded to test the roles of BICD1 and BICD2 in bacterial movement. Initial siRNA knockdowns led to only minor defects in *O. tsutsugamushi* accumulation near the nucleus (Supplementary Fig. 6F, G). We hypothesised this was the result of low levels of remaining BICD1/2 (Supplementary Fig. 6H) and so generated both BICD1 and BICD2 knockout HeLa cell lines to ensure complete removal of adaptors (Supplementary Fig. 6I). Similar to our mitochondrial relocation assay, knockout of BICD1 had no effect on bacterial transport, whereas removal of BICD2 decreased perinuclear clustering. The strongest effect was observed upon double knockout of both adaptors, which resulted in a complete loss of bacterial accumulation near the nucleus (Fig. 4E, F). This demonstrates BICD1 and BICD2 are sufficient to account for bacterial accumulation next to the nucleus. Overall, our in vitro and cell-based findings point toward a model whereby membrane-bound ScaC actively promotes bacterial transport by opening up BICD adaptors for dynein and dynactin recruitment (Fig. 5).

## Discussion

Many cytosolic bacteria, including *Listeria* and certain *Rickettsia* species, move intracellularly by polymerizing actin tails on their surface[23,42–44]. Despite belonging to the Rickettsiaceae family, *O. tsutsugamushi* lacks this ability[2], and has instead developed a mechanism to directly engage the dynein machinery for intracellular transport[2] (Fig. 5). Microtubule-based movement has been observed for another cytosolic bacterium, *Actinobacillus actinomycetemcomitans*, which spreads to adjacent cells via microtubule-containing protrusions[45]. However, *A. actinomycetemcomitans* motility is directed towards the plus ends of microtubules[45] suggesting that it involves kinesin motors. It has also recently been shown that a different cytosolic bacterium,

*Shigella flexneri*, harnesses force from dynein through activating adaptors to drive uncoating of the bacteria-containing vacuole[46].

The exact purpose of dynein-based transport within the *O. tsutsugamushi* life cycle remains enigmatic, especially given our observation that disruption of this transport does not reduce bacterial growth under the measured experimental conditions. It is possible that a growth defect would be observed if the ScaC-BICD1/2 interaction were disrupted under more physiological conditions, such as in the arthropod reservoir, during mammalian infection, or under stress. This is supported by a recent finding from our lab[47] whereby the avirulent strain TA686 is less effective at trafficking to the perinuclear region than virulent strains. There is no growth defect measured in multiple different cell types in vitro, but the TA686 bacteria do not disseminate nor cause disease in vivo. Therefore, we hypothesize that in vivo or under stress, the perinucleus may provide a favourable environment for bacterial growth. This appears to be the case for *Chlamydia trachomatis*, which resides in a vacuole and uses dynein to positions itself near the nucleus to acquire lipids[3,4,48]. Another possibility is that under certain conditions *O. tsutsugamushi* needs to exploit dynein to evade host defences. It is known for example that *Listeria* species use actin-based motility to avoid recognition by the host autophagy machinery which is recruited to damaged vesicles upon phagosome rupture[49]. Given that *O. tsutsugamushi* also breaks out of its vacuole to enter the cytosol, dynein-driven movement could provide a way to quickly retreat from the site of autophagosome formation.

Among the five Sca proteins found in *O. tsutsugamushi* strains, ScaC displays the highest degree of conservation in its passenger domain, underscoring its importance for infection[19]. Previous studies have implicated ScaC in bacterial adhesion to host cells through interaction with the fibronectin receptor, although this process is not sufficient to induce entry on its own[19]. Here, we show that ScaC plays an additional role immediately after cell entry, by directly recruiting the host dynein adaptors BICD2 and BICD1. It was recently shown that during infection of L929 and HUVEC cells ScaC levels are higher in extracellular bacteria than intracellular bacteria[11]. This expression pattern is consistent with ScaC participating in key processes during the early phases of infection, and its later downregulation could provide a mechanism for the bacterium to release dynein and return to the cell periphery for budding after replication has taken place.

In this study we found that knockdown or knockout of BICD1 alone did not produce a phenotype in our cell-based assays, likely due to functional redundancy. We conclude that BICD2 is the predominant adaptor for ScaC in our cell line, although it is possible that BICD1 plays a larger role in other cell types targeted by *O. tsutsugamushi* where it is expressed more abundantly, such as lymphatic endothelial cells[50]. Beyond its physiological roles in the transport of secretory vesicles and maintaining nuclear positioning, BICD2 is reported to be hijacked by viruses such as HIV-1 and HSV-1, which undergo retrograde transport to the host cell nucleus[51–53]. This suggests multiple pathogens have evolved the ability to exploit this specific adaptor, although BICD2 has not previously been implicated in manipulation by a bacterium. We show that ScaC binds the C-terminal CC3 domain of BICD2 to relieve the adaptor from autoinhibition. A similar mechanism had initially been proposed for HIV-1, as purified viral capsids were found to interact with the BICD2 CC3[51]. However, recent reports suggest that HIV-1 binds directly to the dynein tail and is able to hitchhike on dynein/dynactin complexes bound to various adaptors[54]. This approach is remarkably different from the strategy employed by *O. tsutsugamushi*, where association with dynein occurs via a specific cargo-adaptor interaction.

The CC3 domain of BICD2 is known to harbour the binding sites for other conserved host cargoes[31–33]. However, the way ScaC binds BICD2 differs from both RAB6 and RANBP2, reinforcing the notion that BICD2 autoinhibition can be overcome through cargo binding at multiple sites within the CC3 domain[40,41]. Furthermore, ScaC was able

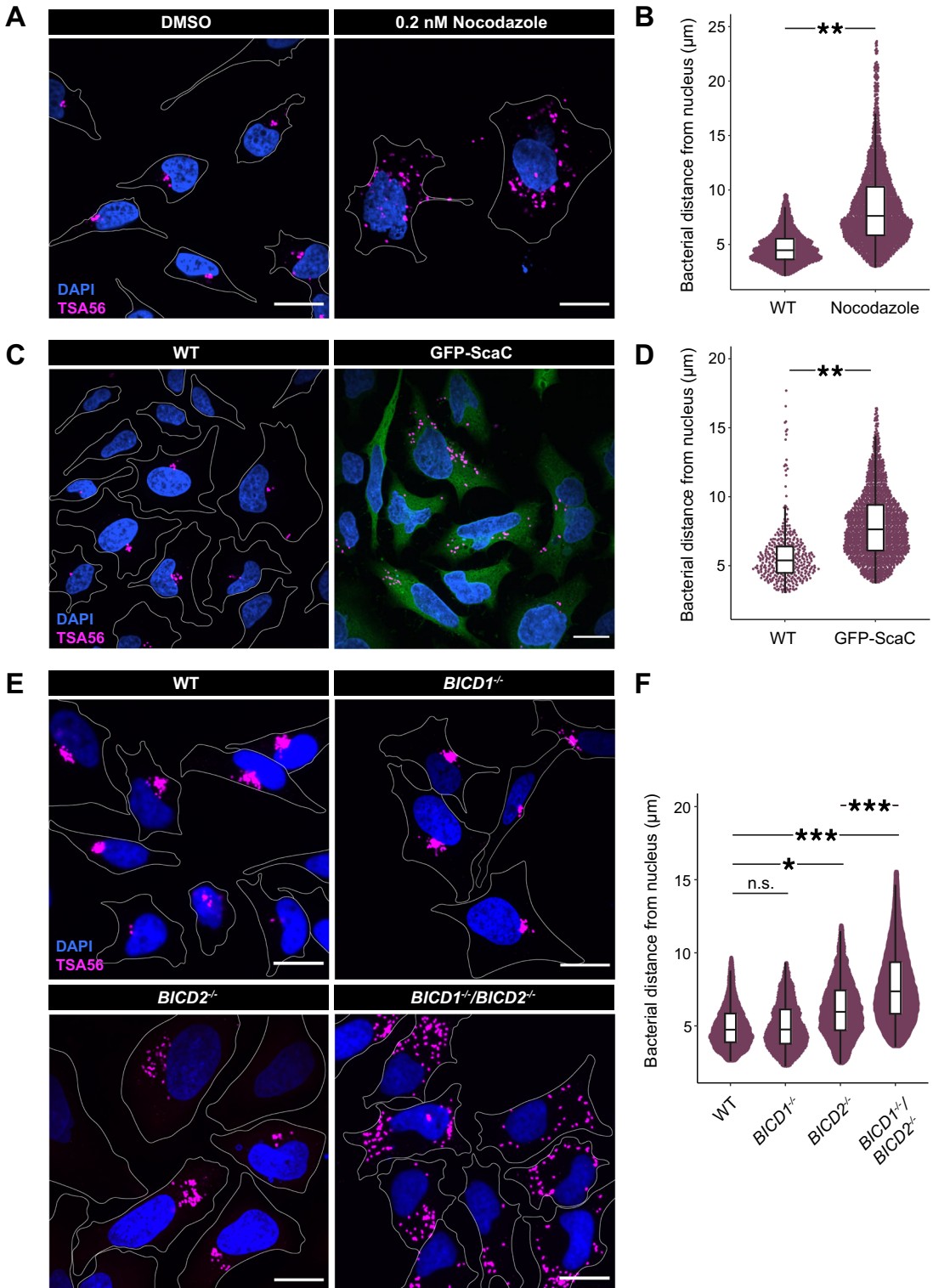

**Fig. 4 | BICD1 and BICD2 are required for perinuclear transport of *O. tsutsugamushi*. A** Representative images showing the localization of *O. tsutsugamushi* strain UT76 (labelled with an antibody against the bacterial surface protein TSA56) at 2 dpi in HeLa ATCC CCL-2 cells treated with either DMSO or 0.2 nM nocodazole. Scale bars indicate 10 μm. **B** Quantification of bacterial distance from the nucleus (μm) in DMSO- and nocodazole-treated samples. The beeswarm plot includes all values between 5th and 95th percentiles. The box plot centre indicates the median, the boxes show the 25th/75th percentile, and whiskers extend to the largest or smallest values within 1.5 × IQR from the hinges. *P*-values were calculated from *n* = 3 independent replicates using a two-tailed *T*-test. \*\**p* < 0.01 (*p* = 0.0095) (**C**) Representative images showing the localization of *O. tsutsugamushi* strain UT76 at 2 dpi in wild-type and GFP-ScaC HeLa ATCC CCL-2 cells. Scale bars indicate 10 μm.

**D** Quantification of bacterial distance from the nucleus in wild-type and GFP-ScaC cell lines. The beeswarm and box plots are as described in (**B**). *P*-values were calculated from *n* = 3 independent replicates using a two-tailed *T*-test. \*\*= *p* < 0.01 (*p* = 0.0012). **E** Representative images showing the localization of *O. tsutsugamushi* strain UT76 at 2 dpi in wild-type, BICD1$^{-/-}$, BICD2$^{-/-}$ or BICD1$^{-/-}$/BICD2$^{-/-}$ HeLa ATCC CCL-2 cells. Scale bars indicate 10 μm. **F** Quantifications of the bacterial distance from the nucleus in each cell line in (**E**). The beeswarm and box plots are as described in (**B**). *P*-values were calculated from *n* = 3 independent replicates using two-tailed ANOVA with a Tukey HSD pot-hoc test. \*$p$ < 0.05, \*\*\*$p$ < 0.001. WT:BICD2$^{-/-}$ $p$ = 0.011, WT:BICD1$^{-/-}$/BICD2$^{-/-}$ $p$ = 4.4 × 10$^{-4}$, BICD2$^{-/-}$:BICD1$^{-/-}$/BICD2$^{-/-}$ $p$ = 1.4 × 10$^{-5}$. Source data are provided as a Source Data file.

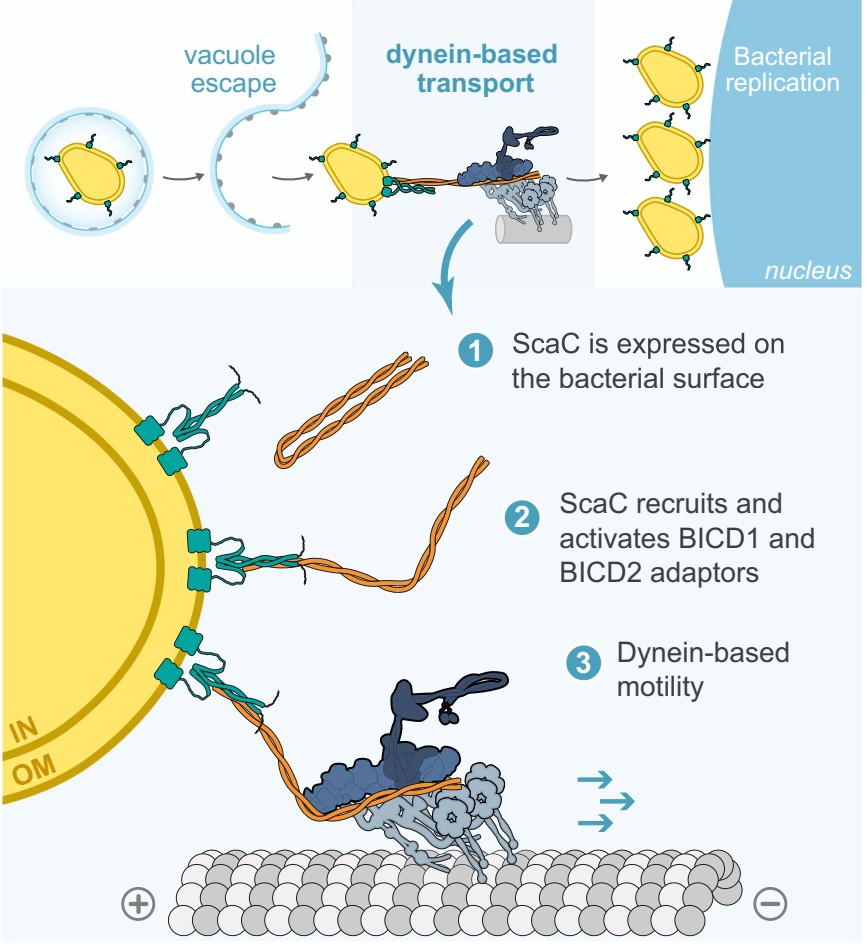

**Fig. 5 | Model of *O. tsutsugamushi* transport by ScaC and BICD adaptors.** (Top panel) Following vacuole escape, the intracellular bacterium *O. tsutsugamushi* undergoes microtubule- and dynein-based transport to reach its replicative niche at the perinucleus. (Bottom panel) We propose a model whereby the autotransporter protein ScaC, which sits on the bacterial outer membrane, recruits the activating adaptors BICD1 and BICD2 to the bacterial surface. We show that binding of ScaC to the C-terminus of BICD2 relieves the adaptor from autoinhibition, thus enabling it to form a complex with dynein and dynactin and initiate movement along microtubules.

to outcompete RANBP2 and, to some extent, RAB6 for BICD2 binding, indicating that it has evolved a higher affinity for BICD2 compared to its host counterparts. This adaptation seems logical, as unlike endogenous cargoes, which require a degree of flexibility to ensure timely and reversible transport, *O. tsutsugamushi* would likely benefit from securing a robust interaction with BICD2 to move efficiently within its host cell. We found that the soluble domain of ScaC forms a coiled coil in vitro, suggesting a mechanism where the passenger domains of two neighbouring ScaC molecules interact with each other after translocation across the outer membrane. We show that ScaC and BICD2 bind with 2:2 stoichiometry, similar to other BICD2-cargo complexes[31,37,38,55]. In the case of RAB6, it is thought that two monomers bind symmetrically on both faces of BICD2[31], whereas RANBP2 has been suggested to bind as a dimer to a single site on BICD2[41]. The latter model could be explained by a high degree of asymmetry in the CC3 domain which would cause the two faces of BICD2 to be structurally distinct, as revealed by a crystal structure of BicD (the *Drosophila* homolog of BICD2)[31]. The dimeric nature of ScaC suggests that is also binds asymmetrically on one side of BICD2. However, it is also possible that the ScaC coiled coil undergoes partial or complete dissociation, allowing each monomer to independently bind a face of BICD2.

Taken together, our findings describe a pathway for bacterial transport in molecular detail and lay the groundwork for further research into how perinuclear transport affects the *O. tsutsugamushi* life cycle. In addition, the discovery of a previously undescribed effector that opens BICD2 through a mechanism distinct from Rab6 and RanBP2 provides an experimental tool for studying BICD1/2 function as well as inducing controlled interference of cellular processes requiring these proteins.

## Methods

### Constructs and plasmids

The full-length mouse BICD2 sequence (UniProt Q921C5) was codon optimized for expression in Sf9 insect cells (Epoch Life Sciences) and cloned into a pACEBac1 vector encoding a N-terminal Strep-tag followed a PreScission (Psc) protease cleavage site (pACEBac1-Strep-Psc). BICD2 truncations 1–400 (BICD2N), 342–800 (BICD2C), 342–541 (BICD2$^{342–541}$), 711–800 (BICD2$^{711-800}$) and 1–710 (BICD2$^{1–710}$) were subcloned into pACEBac1-Strep-Psc. BICD2-SNAP was cloned by insertion of a SNAP tag to the C-terminus of full-length BICD2 in a pACEBac1-Strep-Psc backbone. L742A and L786A mutations were inserted into pACEBac1-Strep-Psc-BICD2$^{711-800}$ (either individually or in combination) by site-directed mutagenesis using the Q5 Site-Directed Mutagenesis Kit (NEB). The full-length mouse BICD1 sequence (UniProt Q8BR07) was synthesized by Twist Biosciences, and residues 713–802 were subcloned into pACEBac1-Strep-Psc.

The passenger domain sequences of ScaA (residues 25–1216), ScaC (residues 33–223), ScaD (residues 30–700) and ScaE (residues 49–494) were synthesized by Epoch Life Sciences and cloned into a pCDNA5/FRT/TO vector fused to a N-terminal eGFP tag. GFP$^{MTS}$ and GFP-BICD2N$^{MTS}$ plasmids were previously described[28]. GFP-ScaC$^{MTS}$ was cloned by addition of a C-terminal MTS tag to the pCDNA5/FRT/TO-eGFP-ScaC plasmid. For bacterial expression of ScaC, the passenger domain was subcloned into a pET22b vector encoding a C-terminal TEV-6His cassette. To generate a ScaC-SNAP construct, a SNAP tag was fused to the C-terminus of ScaC in pET22b-ScaC-TEV-6His. ScaC$^{50-159}$ was subcloned into a pET28 vector encoding a N-terminal Strep-Psc cassette. For bacterial expression of ScaE, the passenger domain was subcloned into a pET28 vector encoding a N-terminal Strep-Psc cassette.

The full-length human RANBP2 sequence was synthesized by Epoch Life Sciences. Residues 2148–2240 fused to a N-terminal GCN4 dimerization tag were subcloned into a pACEBac1 vector with a N-terminal 6×His-ZZ tag followed by a TEV cleavage site. The human RAB6$^{Q72L}$ was subcloned from Addgene plasmid #49483 into a bacteria expression pGEX-4T-1 vector encoding a N-terminal GST tag followed by a TEV cleavage site.

Full-length human LIS1 was expressed from a previously described pFastbac-ZZ-TEV-LIS1 plasmid[28]. For expression of SNAP-tagged full-length human dynein, a previously described pDyn3 plasmid (Addgene #64069) was used[15].

## Mammalian cell lines

Pull down and mitochondrial relocation assays used Flp-In T-REX HeLa (R71407−Thermofisher) derived cell lines. Lentiviruses were produced in Lenti-X 293Ts (NC9834960 - ThermoFisher). Both these cell lines were maintained in Dulbecco's modified Eagle's medium (DMEM) supplemented with 10% FBS and 1% penicillin/streptomycin. Live imaging experiments and *O. tsutsugamushi* propagation was carried out in murine L929 cells (ATCC CCL-1 https://www.atcc.org/products/ccl-1), whilst *O. tsutsugamushi* localization studies used HeLa ATCC CCL-2 cells (ATCC - https://www.atcc.org/products/ccl-2). Both L929 and HeLa cells were cultured in DMEM (41965-039, Gibco) supplemented with 10% FBS. All cells were maintained at 37 °C and 5% $CO_2$ atmosphere. Cell lines were purchased directly from the vendors, but were not further authenticated. All cell lines were tested for mycoplasma.

## Generation of stable expression and knockout cell lines

Stable expression HeLa Flp-In cell lines (GFP, GFP-ScaA, GFP-ScaC, GFP-ScaD, GFP-ScaE, GFP$^{MTS}$, GFP-BICD2N$^{MTS}$ and GFP-ScaC$^{MTS}$) were generated using the FLP-FRT recombination system[28]. In brief, Flp-In T-REX HeLa cells were transfected with 1.8 µg of pOGG4 plasmid and 200 ng of pcDNA5/FRT/TO plasmid encoding the relevant transgene cassette using FuGENE HD transfection reagent (Promega). Cells with a stably integrated transgene were selected with 150 µg/mL hygromycin for 4−5 days. Cells were maintained in DMEM (Gibco) supplemented with 10% FBS and 1% penicillin streptomycin. Transgene expression was induced by adding 10 µg/mL tetracycline to the culture medium.

For overexpression experiments HeLa ATCC CCL-2 cells were transduced for 48 h with a lentivirus encoding a GFP-ScaC cassette. A titration was performed using between 1 and 20 µl of the lentiviral particle stock to a 24 well dish of cells to identify the optimal amount of virus to use to obtain expression without cell death. The GFP-ScaC lentiviral particles were made by co-transfecting the GFP-ScaC cassette (GeneCopoeia), packaging (psPAX2, Addgene #12260 and envelope plasmids (pMD2.G, Addgene #12259) into Lenti-X 293 T cells (with Lipofectamine 3000 (Thermo Fisher). Medium was collected 48 h after transfection and then again 24 h later. Medium containing lentiviral particles was concentrated using a Lenti-X concentrator (Takara,

Japan) and resuspended in Opti-MEM (Thermo Fisher). Viral particles were stored at −80 °C until needed.

Knockout Hela ATCC CCL-2 cell lines were made using a 4D-Nucleofector X (Lonza) and the SE Cell line 4D-Nucleofector X Kit (Lonza) according to manufacturer instructions. In brief, 6 µL of 30 µM sgRNA solution (Synthego) and 1 µL of Cas9 protein (IDT) were diluted in 18 µL SE buffer and incubated at room temperature for 10 min. Approximately $2 \times 10^5$ HeLa ATCC CCL-2 cells were resuspended in 5 µL of SE buffer and added to the sgRNA-Cas9 mix. The sample was then transferred to a nucleofection cuvette. Nucleofection was performed using the SE Buffer setting and pulse code CA123. The cells were recovered in warm DMEM medium for 10 min and plated in a 10-cm dish. For generation of a double BICD1$^{-/-}$/BICD2$^{-/-}$ line, BICD2$^{-/-}$ cells were nucleofected with a BICD1 sgRNA for sequential gene knockout. For genotyping, single colonies were picked and expanded for genomic DNA extraction with the DNeasy kit (Qiagen), followed by PCR amplification of the target genomic region and Sanger Sequencing (Source Bioscience). The sgRNA used were: BICD1 (TGAGTTAGTGACATTCAGTG) and BICD2 (GACGTCGCCGGCGAGACACA). The primers used were: BICD1-F: GAATGCAAATACAGGGTGGCAG, BICD1-R: GGGGTCTTAGTTGGACTTGG, BICD2-F: CGTGACAGCCATG AGGATG, BICD2-R: CGGTAGTAGTCCAGCATGACAC.

## Protein expression

For bacterial expression of recombinant proteins, SoluBL21 *E. coli* cells transformed with the expression plasmid were grown to O.D. 0.6-0.8 and induced with 500 mM IPTG. Expression was carried out at 25 °C for 14 h in a shaking bacterial incubator (speed 180 rpm). Pellets were harvested by centrifugation in a JLA 8.1000 rotor (Beckman Coulter) at 4,000 × g for 20 min at 4 °C, followed by flash-freezing and long-term storage at −80 °C.

For insect cell expression, protein constructs were cloned in pACEBac1 vectors and transformed into EmBacY *E. coli* cells. 4 µg of purified bacmid DNA were transfected into $1 \times 10^6$ Sf9 cells in a well of a 6-well plate using 6 µL of FuGENE HD transfection reagent (Promega). After 7 days of incubation at 27 °C, 1 mL of baculovirus-containing medium was used to infect 50 mL of Sf9 cells at a density of $1 \times 10^6$ cells/mL to generate P2 virus. P2 cultures were incubated at 27 °C for 5 days in a shaking incubator. The P2 virus was harvested by collecting the supernatant after centrifugation at 3,000 × g in an Eppendorf swinging bucket centrifuge (5810 R) for 10 min and was stored at 4 °C. For expression, 10 mL of P2 virus were used to infect 1 L of Sf9 cells at a density of 1.5-2 × $10^6$ cells/mL for 72 h at 27 °C in a shaking incubator. Pellets were harvested by centrifugation at 4,000 × g in an Eppendorf swinging bucket centrifuge (5810 R) for 20 min at 4 °C, followed by flash-freezing and long-term storage at −80 °C.

## Protein purification

All BICD2 and BICD1 constructs were expressed in Sf9 insect cells. Pellets from 1 L of expression culture were resuspended in 45 mL lysis buffer (30 mM HEPES pH 7.2, 300 mM NaCl, 1 mM DTT, 2 mM PMSF) supplemented with 1 cOmplete-EDTA protease-inhibitor tablet (Sigma-Aldrich) and lysed using a Dounce tissue homogenizer with 20-25 strokes. Lysates were cleared by centrifugation in a Type 70 Ti rotor (Beckman Coulter) at 125,000 × g for 45 min at 4 °C followed by filtering with a GF filter (Sartorius). Cleared lysates were then incubated with 1.5 mL of StrepTactin Sepharose beads (IBA Lifesciences) for 1 h at 4 °C on a rotator. Beads were washed on a gravity column with 200 mL wash buffer (30 mM HEPES pH 7.2, 300 mM NaCl, 1 mM DTT), and Strep-tagged proteins were eluted with 5−7 mL of wash buffer supplemented with 3 mM desthiobiotin. For AF488-labelled BICD2-SNAP, beads were incubated with 10 µM SNAP-Surface Alexa Fluor 488 (NEB) for 1 h at 4 °C prior to elution. Eluted proteins were concentrated to a volume of ~0.5 mL and loaded onto a Superose 6 Increase 10/300 column (Cytiva) pre-equilibrated in GF150 buffer (25 mM HEPES pH

7.5, 150 mM KCl, 1 mM MgCl₂, 1 mM DTT). Peak fractions were pooled and concentrated to 1.5-6 mg/mL, depending on the specific construct. Concentrated proteins were flash-frozen and stored at −80 °C.

For purification of GCN4-RANBP2^min, a pellet from 1 L of Sf9 insect cell expression culture was resuspended in 45 mL lysis buffer (200 mM Tris-HCl pH 8, 300 mM NaCl, 20 mM Imidazole, 1 mM DTT, 2 mM PMSF). supplemented with 1 cOmplete-EDTA protease-inhibitor tablet (Sigma-Aldrich) and lysed using a Dounce tissue homogenizer with 20-25 strokes. The lysate was cleared by centrifugation in a Type 70 Ti rotor (Beckman Coulter) at 125,000 × g for 45 min at 4 °C followed by filtering with a GF filter (Sartorius). The cleared lysate was incubated with 2 mL Ni-NTA agarose (Qiagen) on a rotator for 1 h at 4 °C. Beads were washed on a gravity column with 200 mL wash buffer (20 mM Tris-HCl pH 8, 300 mM NaCl, 20 mM Imidazole, 1 mM DTT), and His-tagged proteins were eluted with 5-7 mL of elution buffer (20 mM Tris-HCl pH 8, 300 mM NaCl, 200 mM Imidazole, 1 mM DTT). The eluted proteins were dialysed overnight in 20 mM Tris-HCl buffer (pH 8) supplemented with 1 mM DTT using a 10 kDa MWCO SnakeSkin dialysis tube (ThermoFisher), then concentrated to -0.5 mL and loaded onto a Superose 6 Increase 10/300 column pre-equilibrated in GF150 buffer. Peak fractions were pooled and concentrated to 7.5 mg/mL, flash-frozen and stored at −80 °C.

ScaC, ScaC-SNAP and ScaC^50-159 and ScaE were expressed in SoluBL21 E. coli cells. Pellets from 1 L of expression culture were resuspended in 45 mL lysis buffer (20 mM Tris-HCl pH 8, 300 mM NaCl, 20 mM Imidazole, 1 mM DTT, 2 mM PMSF, 0.3% Triton-X-100) supplemented with 1 cOmplete-EDTA protease-inhibitor tablet (Sigma-Aldrich) and lysed by sonication (3 min, 70% amplitude, pulse: 3 s on, 7 s off) at 4 °C. Lysates were cleared by centrifugation in a Type 70 Ti rotor (Beckman Coulter) at 125,000 × g 30 min at 4 °C followed by filtering with a GF filter (Sartorius). For purification of ScaC and ScaC-SNAP, cleared lysates were incubated with 2 mL Ni-NTA agarose (Qiagen) on a rotator for 1 h at 4 °C. Beads were washed on a gravity column with 200 mL wash buffer (20 mM Tris-HCl pH 8, 300 mM NaCl, 20 mM Imidazole, 1 mM DTT, 10 mM MgCl₂, 1 mM ATP), and His-tagged proteins were eluted with 5-7 mL of elution buffer (20 mM Tris-HCl pH 8, 300 mM NaCl, 200 mM Imidazole, 1 mM DTT). The eluted proteins were dialysed overnight in 20 mM Tris-HCl buffer (pH 8) supplemented with 1 mM DTT using a 10 kDa MWCO SnakeSkin dialysis tube (ThermoFisher) and loaded onto a MonoQ 5/50 GL column for anion exchange chromatography. For purification of ScaE and ScaC^50-159, the cleared lysate was incubated with 1.5 mL StrepTactin Sepharose beads (IBA Lifesciences) for 1 h at 4 °C on a rotator. Beads were washed on a gravity column with 200 mL wash buffer (30 mM HEPES pH 7.2, 300 mM NaCl, 1 mM DTT, 10 mM MgCl₂, 1 mM ATP), and Strep-tagged proteins were eluted with 5-7 mL of wash buffer supplemented with 3 mM desthiobiotin. For all ScaC and ScaE constructs, peak fractions were concentrated to -0.5 mL and loaded onto a Superdex 200 Increase 10/300 column pre-equilibrated in GF150 buffer. Peak fractions were pooled and concentrated to 2-3 mg/mL, flash-frozen and stored at −80 °C. For AF647-labelled ScaC-SNAP, the protein eluted from MonoQ 5/50 GL was incubated with 10 µM SNAP-Surface Alexa Fluor 647 (NEB) for 1 h at 4 °C prior to gel filtration.

For purification of RAB6^Q72L, a pellet from 1 L of SoluBL21 E. coli cell expression culture was resuspended in 45 mL lysis buffer (PBS, 5 mM DTT, 1 mM PMSF, 0.5% Triton-X 100, 10% glycerol, 10 µM GTP) and lysed by sonication (70% amplitude, 2 min, pulse: 2 s on 12 s off) at 4 °C. The lysate was cleared by centrifugation in a Type 70 Ti rotor (Beckman Coulter) at 125,000 × g for 30 min at 4 °C followed by filtering with a GF filter (Sartorius). The cleared lysate was incubated with 3 mL of Glutathione Sepharose 4B resin (Cytiva) supplemented with 10 µM GTP at 4 °C on a rotator for 2 h. The resin was washed with 450 mL of PBS with 10% glycerol and incubated overnight with 35 µL (35 units) thrombin solution at room temperature for cleavage of the GST tag.

Cleaved proteins were separated from the resin using a gravity column, concentrated to -0.5 mL and loaded onto a Superose 6 Increase 10/300 column pre-equilibrated in GF150 buffer supplemented with 10 µM GTP. Peak fractions were pooled and concentrated to 7 mg/mL, flash-frozen and stored at −80 °C.

Dynein was purified from Sf9 insect cells[15]. A pellet from 1 L of expression culture was resuspended in 45 mL lysis buffer (50 mM HEPES pH 7.2, 100 mM NaCl, 10% glycerol, 1 mM DTT and 100 µM ATP, 2 mM PMSF) supplemented with 1 cOmplete-EDTA protease-inhibitor tablet (Sigma-Aldrich) and lysed using a Dounce tissue homogenizer with 30 strokes. The lysate was cleared by centrifugation in a Type 70 Ti Rotor (Beckman Coulter) at 360,000 × g for 45 min in a at 4 ˚C. The cleared lysate was incubated with 4 mL of IgG beads (Cytiva) at 4 °C on a rotator for 4 h. The beads were washed with 150 mL wash buffer (50 mM HEPES pH 7.2, 100 mM NaCl, 10% glycerol, 1 mM DTT and 100 µM ATP) and incubated with 10 µM SNAP-Cell TMR-Star dye (NEB) for 1 h at 4 °C. The beads were then transferred to a gravity column and washed with 100 mL of TEV Buffer (50 mM Tris-HCl pH 7.4, 150 mM KAc, 2 mM MgAc, 1 mM EGTA, 10% (v/v) glycerol, 0.1 mM ATP, 1 mM DTT). The protein was cleaved from the beads by overnight incubation with 400 µg of TEV protease at 4 °C. The flowthrough containing the cleaved protein was collected and concentrated to -2 mg/mL, then loaded onto a TSKgel G4000SWXL column pre-equilibrated with GF150 buffer supplemented with 0.1 mM ATP. Peak fractions were pooled and concentrated to 2 mg/mL. The concentrated protein was supplemented with 10% glycerol, flash-frozen and stored at −80 °C.

Dynactin was purified natively from pig brains[15]. Brains were purchased from a butcher within 2 h of slaughter and brought back to the lab in ice-cold PBS. They were cleaned in homogenization buffer (35 mM PIPES pH 7.2, 5 mM MgSO₄, 100 µM EGTA and 50 µM EDTA), squashed as thinly as possible in a plastic bag and flash frozen in liquid nitrogen. Two frozen brains were homogenised in a blender and resuspended in homogenization buffer supplemented with 1.6 mM PMSF, 1 mM DTT and four complete-EDTA protease-inhibitor tablets per 500 ml (Roche). The lysate was cleared first in a JLA 16.250 (Beckman Coulter) at 26,000 × g for 15 min at 4 °C, then in a Type 45 Ti rotor (Beckman Coulter) at 150,000 × g for 50 min at 4 °C. The supernatant was filtered through a glass fibre filter (Sartorius), followed by a 0.45-µm filter (Elkay Labs), and loaded on a column packed with 250 ml of SP-Sepharose (Cytiva) equilibrated with SP buffer (35 mM PIPES pH 7.2, 5 mM MgSO₄, 1 mM EGTA, 0.5 mM EDTA, 1 mM DTT and 0.1 mM ATP) using an Akta Pure (Cytiva). The column was washed with SP buffer supplemented with 3 mM KCl, then eluted via a linear gradient up to 250 mM KCl over 3 column volumes. The peak containing dynactin was filtered in a 0.22-µm filter (Elkay Labs) and loaded on a MonoQ 16/10 column (Cytiva) pre-equilibrated with MonoQ buffer (35 mM PIPES, pH 7.2, 5 mM MgSO₄, 100 µM EGTA, 50 µM EDTA and 1 mM DTT). The column was washed with MonoQ buffer and the dynactin eluted by a linear gradient up to 150 mM KCl over 1 column volume, followed by another linear gradient up to 350 mM KCl over 10 column volumes. The peak containing dynactin was pooled and concentrated to ~ 3 mg ml⁻¹ and loaded on a size exclusion column (TSKgel G4000SW^XL - Tosoh Bioscience) pre-equilibrated with GF150 buffer (25 mM HEPES pH 7.2, 150 mM KCl and 1 mM MgCl₂) supplemented with 5 mM DTT and 0.1 mM ATP. The peak containing dynactin was pooled and concentrated to -3 mg ml⁻¹. Aliquots (3 µl) were flash frozen in liquid nitrogen and stored at −80 °C.

Lis1 was expressed in Sf9 cells[28]. A cell pellet from 1 litre expression was resuspended in 50 ml lysis buffer B [50 mM Tris-HCl pH 8.0, 250 mM KAc, 2 mM MgAc2, 1 mM EGTA, 10% (v/v) glycerol, 0.1 mM ATP, 1 mM DTT, 2 mM PMSF] and lysed with -20 strokes in a 40-ml dounce tissue grinder (Wheaton). The lysate was clarified at 150,000 × g for 30 min at 4 °C in a Type 45 Ti Rotor (Beckman Coulter) and supernatant added to 3 ml IgG Sepharose 6 Fast Flow beads

(Cytiva) pre-equilibrated with lysis buffer B for 4 h at 4 °C. The beads were washed with 150 ml of lysis buffer B in a gravity flow column. The beads were transferred to a 5 ml, which was filled completely with buffer B before addition of 400 μg of TEV protease. After overnight cleavage at 4 °C the Lis1 was collected by using a gravity flow column, concentrated to between 5 mg/ml and run over a Superdex 200 Increase 10/300 (Cytiva) column equilibrated with GF150 buffer supplemented with 5 mM DTT and 10% glycerol. Peak fractions were pooled and concentrated to ~5 mg/ml, aliquoted, flash frozen and stored at −80 °C.

## Co-IP mass spectrometry
HeLa Flp-In cells expressing either GFP, GFP-ScaA, GFP-ScaC, GFP-ScaD or GFP-ScaE transgenes were grown to confluence in a 15 cm tissue culture dish in the presence of 10 μg/mL tetracycline. Cells were detached using 0.25% trypsin-EDTA solution (ThermoFisher) and pellets were washed twice with PBS. Pellets were lysed with co-IP lysis buffer (10 mM Tris pH 7.5, 150 mM NaCl, 0.5 mM EDTA, 0.5% NP-40, protease inhibitor, 1 mM PMSF) on ice for 30 min, and lysates were cleared in an Eppendorf 5424 R centrifuge at 21,000 × g at 4 °C for 15 min. Protein concentrations were measured by Bradford assay. Samples were diluted to 4 mg/mL with dilution buffer (10 mM Tris pH 7.5, 150 mM NaCl, 0.5 mM EDTA), and 500 μL were incubated with 25 μL of magnetic GFP-trap beads (Chromotek) for 1 h at 4 °C on a rotator (speed 5 rpm). The beads were washed three times with 500 μL dilution buffer followed by buffer exchange with 100 mM ammonium bicarbonate, which was repeated twice. After the last the wash, a minimum amount of buffer was left to cover the beads. The cysteines were reduced with 30 μL of 10 mM DTT and alkylated with 30 μL of 55 mM iodoacetamide. Proteins were digested on beads with 1 μg trypsin (Promega, UK) for 18 h at 37 °C. Peptides were acidified with the addition of 4 μL formic acid 2% (v/v). The bead/peptide mix was then centrifuged in an Eppendorf 5424 R centrifuge at 14,000 × g for 5 min and 5 μL of supernatant were diluted with 15 μL of 0.1% formic acid. 5 μL of the diluted supernatant was injected for LC-MS/MS analysis. LC-MS/MS was performed on an Ultimate U3000 HPLC (ThermoFisher Scientific) hyphenated to an Orbitrap QExactive HFX mass spectrometer (ThermoFisher Scientific). Peptides were trapped on a C18 Acclaim PepMap 100 (5 μm, 300 μm × 5 mm) trap column (ThermoFisher Scientific) and eluted onto a C18 Acclaim PepMap100 3 μm, 75 μm × 250 mm (ThermoScientific Dionex) using an 85 min gradient of acetonitrile (5 to 35%). For data dependent acquisition, MS1 scans were acquired at a resolution of 120,000 (AGC target of 3e6 ions with a maximum injection time of 25 ms) followed by 25 MS2 scans acquired at a resolution of 30,000 (AGC target of 5e4 ions with a maximum injection time of 150 ms) using a collision induced dissociation energy of 27. Dynamic exclusion of fragmented m/z values was set to 50 s. Raw data were imported and processed in Proteome Discoverer v3.1 (Thermo Fisher Scientific). The raw files were submitted to a database search using Proteome Discoverer with Sequest HT against the UniProt reference proteome for Homo sapiens. The processing step consisted of a double iterative search using INFERIS Rescoring algorithm on a first pass with methionine oxidation and cysteine carbamidomethylation set as variable and fixed modifications, respectively. For the second pass, all spectra with a confidence filter worse than "high" were researched with Sequest HT including additional common protein variable modifications (Deamidated (N,Q); gln to pyro-Glu (Q); N-terminal acetylation and Methionine loss). The spectra identification was performed with the following parameters: MS accuracy, 10 p.p.m.; MS/MS accuracy of 0.02 Da; up to two trypsin missed cleavage sites were allowed. Percolator node was used for false discovery rate estimation and only rank 1 peptide identifications of high confidence (FDR < 1%) were accepted. Three biological replicates were performed for each sample.

## Mitochondria relocation assay
HeLa GFP[MTS], GFP-BICD2N[MTS] or GFP-ScaC[MTS] cells were treated with 10 μg/mL tetracycline overnight to induce transgene expression. Staining was performed with 20 μM Hoechst (ThermoFisher), 5 μg/mL CellMask Orange (ThermoFisher) and 50 nM MitoTracker Deep Red (ThermoFisher) for 20 min at 37 °C. Samples were imaged using a Nikon Ti2 inverted fluorescence microscope with a 20x/0.75NA Air lens using the NIS Elements software (version 5.1). For the mitochondria relocation assay with nocodazole, GFP-ScaC[MTS] cells were treated with either DMSO or 10 μM nocodazole for 3 h at 37 °C prior to imaging. For the mitochondrial relocation assay with siRNA knockdowns, GFP[MTS] and GFP-ScaC[MTS] cells were transfected with 10 nM of either a non-targeting (scrambled) siRNA (Silencer Select Negative Control #1 siRNA, Invitrogen) or siRNAs directed against BICD1 (Invitrogen, ID: 147217) or BICD2 (Invitrogen, ID: 136849) using Lipofectamine RNAi-Max (ThermoFisher) according to manufacturer instructions. In some conditions, siRNAs against BICD1 and BICD2 were used in combination, for a final siRNA concentration of 20 nM. Imaging was performed 72 h after siRNA transfection. Three biological replicates were performed for each sample and condition. Mitochondrial spread relative to the nucleus was quantified in Fiji ImageJ (version 2.8.0) using a publicly available ImageJ script (https://github.com/jboulanger/imagej-macro/tree/main/Cell_Organization)[22,28]. The mitochondrial regions, the nucleus and the cell border were defined by segmentation of MitoTracker, Hoechst and CellMask signals, respectively. The mitochondrial spread was measured as the trace of the second moment matrix of the MitoTracker signal intensity in each cell, akin to a standard deviation in 2D.

## Growth quantification of ScaC-GFP cells
GFP or GFP-ScaC HeLa Flp-In cells were seeded in a 12-well dish at a density of $5 \times 10^4$ cells/well in DMEM medium supplemented with 10 μg/mL tetracycline. The total cell count was determined at 1, 2, 3 and 4 days post-seeding using a Countess 3 Cell Counter (ThermoFisher).

## Immunofluorescence analysis of BICD1 and BICD2
HeLa GFP[MTS] and GFP-ScaC[MTS] cells were fixed with 4% PFA in PBS for 10 min at room temperature followed by permeabilization with 0.1% Triton X-100 in PBS for 10 min at room temperature. Blocking was carried out with 2% BSA in PBS with 0.1% Tween 20 for 1 h at room temperature. Samples were incubated with anti-BICD1 (Atlas Antibodies, HPA041309, 1:200 dilution) or anti-BICD2 (Atlas Antibodies, HPA023013, 1:200 dilution) antibodies diluted in 1% BSA in PBS solution overnight at 4 °C. Secondary antibody incubation was carried out for 1 h at room temperature with GFP-Booster Alexa Fluor 488 (Chromotek, gb2AF488, 1:500 dilution) and Alexa Fluor 647 donkey anti-rabbit IgG (ThermoFisher, A-31573, 1:1000 dilution) diluted in 1% BSA in PBS. Samples were mounted using Mounting Medium with DAPI (Abcam, ab104139) and imaged with a Nikon W1 Spinning Disk inverted microscope using a 100x/1.4NA Oil lens using Nikon Elements software (version 5.4). Three biological replicates were performed for each sample. Colocalization of BICD1 and BICD2 with GFP[MTS] and GFP-ScaC[MTS] was quantified using a custom-made python script. Cells were segmented using Cellpose[56]. After applying a gaussian filter to the signals for BICD1/2 and GFP[MTS]/GFP-ScaC[MTS], colocalization was quantified based on the Mander's coefficient (M1), which measures the fraction of the total BICD1/2 fluorescent signal that overlaps with fluorescence in the GFP[MTS]/GFP-ScaC[MTS] channel.

## qPCR analysis of BICD1 and BICD2 expression
Total RNA was extracted from GFP-ScaC[MTS] and HeLa ATCC CCL-2 cells using the RNeasy RNA extraction kit (Qiagen) according to manufacturer instructions. cDNA was synthesized using the ProtoScript First Stand cDNA Synthesis Kit (NEB). The obtained cDNA was diluted 10-

fold to 200 μL with nuclease-free water. qPCR reactions were assembled using 5 μL of Fast SYBR Green Master Mix (ThermoFisher), 1 μL of each primer (final concentration 200 nM), 1 μL of diluted cDNA and nuclease-free water for a final volume of 10 μL. Three biological replicates were performed, each including two technical duplicates. The primers used were: BICD1-F (CTTGTCTTTGTCAGGTAGCC), BICD1-R (CCCATGTCCGCTCAATAAG), BICD2-F (GACGGAGCTGAAGCAGTTG), BICD2-R (CCGGAATTTGTACTCCTTGATGTC), GAPDH-F (TTGGCTACAGCAACAGGGTG) and GAPDH-R (GGGGAGATTCAGTGTGGTGG). For quantification, we used the log2(-ΔCt) method normalizing BICD1 and BICD2 expression to GAPDH expression.

## Bacterial propagation and quantitation

The *O. tsutsugamushi* strain UT76 was propagated in L929 cells using 25- or 75-cm² flasks. Cells were grown in DMEM (41965-039, Gibco) at 37 °C (uninfected) or 35 °C (uninfected) and 5% CO₂ atmosphere. The growth media was supplemented with 10% FBS in uninfected cells and 5% FBS in infected cells. Cells were infected when they were 60–90% confluent using frozen stocks of bacteria. At 6 days post infection the supernatant was removed, cells washed once with media, then the cells resuspended in fresh media. Cells were detached by mechanical detachment using a cell scraper (Corning, ID:3010) and transferred to an Eppendorf tube. Host cells were lysed with 0.1 mm glass beads (Sigma-Aldrich, ID:11079101) using a mini homogenizer (Fisher Scientific, ID:15525799) at power XXX for 1 min. Lysed cells were centrifuged at $800 \times g$ for 3 mins to pellet beads and host cell debris and supernatant transferred to a new Eppendorf tube. Bacteria were centrifuged at $14,000 \times g$ for 5 mins. Supernatant was discarded and the pellet resuspended in SPG buffer (0.218 M sucrose, 3.76 mM KH₂PO₄, 7.1 mM KH₂PO₄, 4.9 mM monosodium L-glutamic acid). Bacterial stocks were quantified by qPCR and frozen at −80 °C until use. For infection of HeLa cells, frozen bacteria were used to infect the cells directly with MOI 50:1. This MOI was determined based on the bacterial copy number and the host cell copy number as measured by qPCR and cell counter respectively.

## Bacterial infection for microscopy

Wild-type, GFP-ScaC-expressing, BICD1⁻/⁻, BICD2⁻/⁻, or double BICD1⁻/⁻/BICD2⁻/⁻ HeLa cells (ATCC) were seeded into sterile μ-Slide 8-well ibiTreat chambers (Ibidi) at a density of $1 \times 10^4$ cells per well and incubated overnight at 37 °C in a humidified 5% CO₂ atmosphere. Cells were infected with *O. tsutsugamushi* at a multiplicity of infection (MOI) of 300:1 for 3 h, washed three times with serum-free medium to remove unbound bacteria, and incubated for an additional 48 h under the same conditions.

For nocodazole-treated infections, HeLa cells were seeded as above and pretreated with 0.2 nM nocodazole for 30 minmin prior to infection with the bacteria at an MOI of 100:1. Following a 3-h infection period, cells were washed three times with serum-free medium and incubated for a further 48 h in the presence of 0.2 nM nocodazole.

For experiments using siRNA-treated cells, HeLa cells were seeded into sterile μ-Slide 8-well ibiTreat dishes (Ibidi) at a density of $1 \times 10^4$ cells/well and incubated overnight at 37 °C. Cells were transfected with either a scrambled siRNA (Silencer Select Negative Control #1, Invitrogen) or siRNAs targeted against *BICD1* (Invitrogen, ID: 147217) and *BICD2* (Invitrogen, ID: 136849) using Lipofectamine 3000 (Invitrogen) for a final siRNA concentration of 50 nM. After 24 h, the cells were incubated with the bacteria at MOI 50:1 for 3 h, followed by media replacement to remove excess bacteria.

## Immunofluorescence analysis of bacterial localization

At appropriate time points, cells were fixed with 4% formaldehyde for 10 min at room temperature. Cells were permeabilized by incubation with absolute ethanol for 1 h on ice followed by incubation with 0.5% Triton X-100 in PBS for 30 min on ice. Permeabilized cells were washed twice with PBS and incubated with an in-house generated antibody against Tsa56 (rat monoclonal, 1:200 dilution) diluted in PBS containing 1 mg/ml BSA at 4 °C overnight. Cells were then washed twice with PBS and incubated with a goat anti-rat Alexa Fluor 488 antibody (ThermoFisher, A-11006, 1:1000 dilution), DAPI (1:1000) and HCS CellMask deep red (ThermoFisher, 1:5000 dilution) for 30 min at 37 °C in the dark. Cells were maintained in mounting medium (20 mM Tris pH 8.0, 0.5% N-propyl-gallate, 90% glycerol). Imaging was performed using a Zeiss LSM 700 confocal fluorescence microscope with an HBO 100 illuminating system equipped with a 63 × /1.4 Plan-APOCHROMAT objective lens (Carl Zeiss) and 405 nm, 488 nm, and 555 nm laser lines acquired ZEN 2011 SP7 FP3 (version 14.0.0.0) software. The distance between *O. tsutsugamushi* and the nucleus was measured using CellProfiler (version 4.2.6)[57]. In brief, the host cell nucleus was segmented using an adaptive strategy (Otsu's method) with adaptive window 24, in which the cytosol was defined as a secondary object correlated with the nucleus. *O. tsutsugamushi* was segmented using a global Otsu's strategy. The distance between cytosolic *O. tsutsugamushi* (i.e. present in the cytosol and non-overlapping with the nucleus) and the nucleus was determined using the "expand until adjacent" method, in which the closet distance between the nuclear rim and the border of the *O. tsutsugamushi* object is measured.

## Live imaging of *O. tsutsugamushi*

Purified bacteria were labelled with CFSE[58] (catalog number C34554 from LifeTechnologies, USA). 5 mM aliquots of dye dissolved in DMSO were stored in 1 μl volumes at −20 °C. For labelling, one aliquot was thawed to room temperature. This was diluted to a 5 μM working dilution in PBS containing 2% BSA. Purified bacteria were pelleted at $20,238 \times g$ at room temperature, then resuspended in the 5 μM dye solution. Samples were incubated for 15 min at 37 °C for 30 min at room temperature, in the dark. Bacteria were then pelleted and resuspended in RPMI + 5% FBS and incubated at room temperature for 30 min to quench remaining dye. Bacteria were repelleted and resuspended in fresh media, and were then ready for use. CFSE-labelled bacteria were added to growing L929 cells in ibidi chambered cover-glass slides, then imaged immediately by confocal microscopy. Imaging was performed using a Leica TCS SP8 confocal microscope (Leica Microsystems, Germany) using the Leica LAS X software package (version 4).

## In Vitro Growth Rate Analysis of *O. tsutsugamushi*

Wild-type, GFP-ScaC-expressing, BICD1⁻/⁻, BICD2⁻/⁻, or double BICD1⁻/⁻/BICD2⁻/⁻ HeLa cells (ATCC) were seeded overnight at $3 \times 10^4$ cells/well in a 48-well plate. For experiments using siRNA-treated cells, HeLa cells were seeded into 48-well plates at a density of $1 \times 10^4$ cells/well and incubated overnight at 37 °C. Cells were transfected with either a scrambled siRNA (Silencer Select Negative Control #1, Invitrogen) or siRNAs targeted against *BICD1* (Invitrogen, ID: 147217) and *BICD2* (Invitrogen, ID: 136849) using Lipofectamine 3000 (Invitrogen) for a final siRNA concentration of 50 nM for 24 h.

Once the cells were prepared, the medium was removed, and bacteria were added to the cells at a multiplicity of infection (MOI) of 50, followed by incubation for 3 h. After incubation, bacteria were removed, and the infected cells were washed three times with plain medium. The cells were further cultured in growth medium under the same conditions. At specific time points, the medium was removed, and the infected cells were washed three times with PBS. DNA was extracted by adding alkaline lysis buffer (25 mM NaOH, 0.2 mM EDTA) directly into the wells, followed by boiling the plates at 95 °C for 30 min to inactivate the bacteria. The plates were stored at 4 °C until further testing. Bacterial concentration was quantified by qPCR. The primers

and TaqMan probe used for the 47 kDa target gene were as follows: 47 kDa FW (5′-TCCAGAATTAAATGAGAATTTAGGAC-3′), 47 kDa RV (5′-TTAGTAATTACATCTCCAGGAGCAA-3′), and 47 kDa probe (5′-FAM-TTCCACATTGTGCTGCAGATCCTTC-TAMRA-3′). The qPCR mixture consisted of 1X qPCR Probe Mix LO-ROX (PCR Biosystems, cat# PB20.21, UK), 0.1 μM of each forward and reverse primer, 0.2 μM probe, sterile distilled water, and 1 μL of extracted DNA. Real-time PCR was performed on a CFX Duet Thermal Cycler (Bio-Rad, USA) using the following conditions: initial denaturation at 95 °C for 2 min, followed by 45 cycles of 95 °C for 15 s and 60 °C for 30 s, with fluorescence acquisition during the annealing/extension phase. DNA copy numbers were calculated by comparison with a standard curve.

### Size Exclusion Chromatography (SEC) reconstitution assays

For SEC reconstitution experiments, proteins were mixed and incubated on ice for 15 min to allow for complex formation. BICD2 and ScaC constructs were mixed in a 1:2 molar ratio. BICD2N and BICD2C were mixed in a 1:1 molar ratio. Complexes were diluted to a final volume of 50 μL in GF150 buffer and injected into a Superose 6 Increase 3.2/300 column pre-equilibrated with GF150 buffer using a 100 μL loop. Fractions underlying protein peaks were visualized by SDS-PAGE followed by Comassie Blue staining.

### Size-exclusion chromatography with multi-angle light scattering (SEC-MALS)

The molecular masses of BICD2, ScaC and the BICD2:ScaC complex were determined in solution using SEC-MALS. Measurements were performed using a Wyatt Heleos II 18 angle light scattering instrument coupled to a Wyatt Optilab rEX online refractive index detector. Samples at a concentration of 1 mg/mL were resolved on a Superose 6 10/300 analytical gel filtration column coupled to an Agilent 1200 series LC system running at 0.5 ml/min in GF150 buffer before then passing through the light scattering and refractive index detectors in a standard SEC MALS format. Protein concentration was determined from the excess differential refractive index based on 0.186 ΔRI for 1 g/ml or with the sequence-based UV extinction coefficient determined in ProtParam. The measured protein concentration and scattering intensity were used to calculate molecular mass from the intercept of a Debye plot using Zimm's model as implemented in Wyatt's ASTRA software. The experimental setup was verified for each set of runs using a BSA standard run of the same sample volume. The BSA monomer peak was used to check mass determination and to evaluate interdetector delay volumes and band broadening parameters that were subsequently applied during analysis of BICD2, ScaC and BICD2:ScaC.

### Circular dichroism

Measurements were made in GF150 buffer (25 mM HEPES pH 7.2, 150 mM KCl, 1 mM MgCl$_2$, 1 mM DTT) using a Jasco 815 Spectropolarimeter and a 0.1 cm pathlength cuvette. Spectra were recorded at 25 °C as the average of 8 scans using 2 mg/ml concentration of ScaC protein with a scan rate of 50 nm/min, bandwidth 1 nm and corrected for the contribution of buffer by subtraction of a GF150 blank.

### Pulldown assays

For in vitro pull-down of strep-tagged proteins, 200 nM of strep-tagged BICD2[711-800] (either wild-type or L742A, L786A and L742A/L786A mutant versions) were mixed with 700 nM of either GCN4-RANBP2[min], RAB6[Q72L] or ScaC in GF100 buffer (25 mM HEPES pH 7.2, 100 mM KCl, 1 mM DTT) and were incubated on ice for 20 min to allow for complex formation. Complexes were then diluted to 150 μL with GF100 buffer and 5 μL were set aside for input analysis by SDS-PAGE. Samples were incubated with 30 μL of StrepTactin Sepharose beads (IBA Lifesciences) pre-equilibrated in GF100 for 20 min at 4 °C on a rotator

(speed 20 rpm). Beads were washed 5 times with 500 μL of GF150 buffer (25 mM HEPES pH 7.2, 150 mM KCl, 1 mM DTT) by pelleting in an Eppendorf 5424 R centrifuge at 100 × g. Bound fractions were eluted by incubating the beads with 100 μL GF150 supplemented with 3 mM desthiobiotin for 10 min at 4 °C on a rotor (speed 20 rpm). 20 μL of eluate were mixed with 5 μL NuPAGE LDS Sample Buffer (4X) (ThermoFisher), boiled at 95 °C for 10 min and loaded onto a SDS-PAGE gel. Bands were visualized by Comassie Blue staining.

For pulldown assays with THP-1 and MutuDC lysates, $1 \times 10^7$ cells were lysed in buffer co-IP lysis buffer (10 mM Tris pH 7.5, 150 mM NaCl, 0.5 mM EDTA, 0.5% NP-40, protease inhibitor, 1 mM PMSF) on ice for 30 min, and lysates were cleared by centrifugation in an Eppendorf 5424 R centrifuge at 21,000 × g at 4 °C for 15 min. 200 nM of Strep-tagged ScaC or ScaE were mixed with 30 μL of StrepTactin Sepharose beads (IBA Lifesciences) pre-equilibrated in GF100 for 20 min at 4 °C on a rotator (speed 20 rpm). The beads conjugated with the Sca proteins were then incubated with 1 mg of either THP-1 or MutuDC lysate for or 2 h at 4 °C on a rotator (speed 5 rpm). The beads were washed 5 times with 500 μL of GF150 buffer and the bound fractions were eluted by incubating the beads with 100 μL GF150 supplemented with 3 mM desthiobiotin for 10 min at 4 °C on a rotor (speed 20 rpm). The eluates were analysed by western blotting.

### Cross-linking mass spectrometry

50 μg of BICD2:ScaC complex were crosslinked for 60 min at room temperature at a 1:500 EDC ratio and quenched with 50 mM Tris buffer. The quenched solution was reduced with 5 mM DTT and alkylated with 20 mM idoacetamide. An established SP3 protocol[59,60] was used to clean-up and buffer exchange the reduced and alkylated protein. Briefly, proteins were washed with ethanol using magnetic beads for protein capture and binding. The proteins were resuspended in 100 mM NH$_4$HCO$_3$ and digested with trypsin (Promega, UK) at an enzyme-to-substrate ratio of 1:20 and protease max 0.1% (Promega, UK). Digestion was carried out overnight at 37 °C. Clean-up of peptide digests was performed using HyperSep SpinTip P-20 (Thermo Scientific) C18 columns, with 80% Acetonitrile as the elution solvent. Peptides were then evaporated to dryness via Speed Vac Plus (Savant, USA). Dried peptides were suspended in 3% Acetonitrile and 0.1 % formic acid and analysed by nano-scale capillary LC-MS/MS using a Ultimate U3000 HPLC (Thermo Scientific) to deliver a flow of ~250 nL/min. Peptides were trapped on a C18 Acclaim PepMap100 5 μm, 100 μm × 20 mm nanoViper (Thermo Scientific) before separation on a PepMap RSLC C18, 2 μm, 100 A, 75 μm × 50 cm EasySpray column (Thermo Scientific). Peptides were eluted on a 120 min gradient with acetonitrile and interfaced via an EasySpray ionisation source to a quadrupole Orbitrap mass spectrometer (Q-Exactive HFX, ThermoScientific, USA). MS data were acquired in data dependent mode with a Top-25 method. High resolution full mass scans were carried out ($R = 120,000$, $m/z$ 400 – 1550) followed by higher energy collision dissociation (HCD) with stepped collision energy range 26, 30, 34 % normalised collision energy. The tandem mass spectra were recorded ($R = 60,000$, AGC target $= 1 \times 10^5$, maximum IT = 120 ms, isolation window $m/z$ 1.6, dynamic exclusion 50 s). For analysis of cross-linking data, Xcalibur raw files were converted to MGF files using ProteoWizard[61] and cross-links were analysed by MeroX[62]. Searches were performed against a database containing only BICD2 and ScaC sequences to minimise analysis time with a decoy data base based on peptide sequence shuffling/reversing. Search conditions used 3 maximum missed cleavages with a minimum peptide length of 5, cross-linking targeted residues were K, S, T, and Y, cross-linking modification masses were 54.01056 Da and 85.98264 Da. Variable modifications were carbmidomethylation of cysteine (57.02146 Da) and Methionine oxidation (15.99491 Da). False discovery rate was set to 1 %, and assigned cross-linked spectra were manually inspected.

## In vitro TIRF motility assay

Microtubules were made by mixing 4 µg of HiLyte Fluor 488 tubulin (Cytoskeleton), 4 µg of biotinylated tubulin (Cytoskeleton) and 100 µg of unlabelled pig tubulin in BRB80 buffer (80 mM PIPES pH 6.8, 1 mM $MgCl_2$, 1 mM EGTA, 1 mM DTT) for a final volume of 10 µL. 10 µL of polymerization buffer (2X BRB80 buffer, 20% (v/v) DMSO, 2 mM Mg-GTP) were added to the tubulin mix for a total volume of 20 µL. The reaction was incubated on ice for 5 min, followed by incubation at 37 °C for 2 h in the dark. Microtubules were spun at $21,000 \times g$ for 8 min and the supernatant was aspirated to remove non-polymerised tubulin. The microtubule pellet was resuspended in 100 µL Taxol-BRB80 buffer (BRB80 supplemented with 40 µM paclitaxel), followed by another round of centrifugation at $21,000 \times g$ for 8 min. The pellet was resuspended in 60 µL Taxol-BRB80 buffer and microtubule concentration was measured by Bradford assay. Microtubules were diluted to a final concentration of 0.6 mg/mL in Taxol-BRB80 buffer and stored overnight at room temperature in the dark.

The motility chambers were assembled by applying two thin (7-8 mm) strips of double-sided tape (Tesa) on a glass slide, spaced ~4–5 mm apart, and attaching on top a coverslip pre-cleaned with piranha solution. The chambers were incubated with PLL-PEG-Biotin (SuSOS AG) to functionalize the coverslip, followed by two washes with 10 µL TIRF buffer (30 mM HEPES pH 7.2, 5 mM MgSO4, 1 mM EGTA, 2 mM DTT) and a wash with 10 µL streptavidin (NEB). The chamber was further washed with 10 µL TIRF buffer followed by a 1 min incubation with 2 µL of microtubules diluted in 8 µL TIRF buffer supplemented with 50 mM KCl and 1.25 mg/mL casein (TIRF-Casein). Unbound microtubules were washed out with 20 µL TIRF-Casein buffer.

For complex formation, individual proteins were first diluted in TIRF-Casein buffer to the following final concentrations: TMR-dynein to 0.4 mg/mL, dynactin to 0.4 mg/mL, BICD2N to 0.8 mg/mL, BICD2 to 1.2 mg/mL, BICD2-AF488 to 1.5 mg/mL, ScaC to 0.6 mg/mL, ScaC-AF647 to 1.2 mg/mL and Lis1 to 0.4 mg/mL. Complexes were assembled by mixing 1 µL of each diluted protein and incubating on ice for 15 min. Complexes were then diluted to a final volume of 10 µL in TIRF-Casein buffer. 1 µL of diluted complex was added to 19 µL of motility reaction buffer (TIRF-Casein Buffer supplemented with 0.2 mg/mL catalase, 1.5 mg/mL glucose oxidase, 0.45% (w/v) glucose, 1% BME and 5 mM Mg-ATP). 10 µL of the motility reaction were flown into the chamber and the sample was imaged immediately using a TIRF microscope (Nikon Eclipse Ti inverted microscope equipped with a Nikon 100x TIRF oil immersion objective and driven using µManager software version 1.4.22). For single-colour samples, an image of the microtubules was taken using the 488 nm laser, and a 500-frame movie (200 ms exposure) was acquired using the 571 nm laser. For dual-colour imaging of ScaC-AF647 and BICD2-488, movies were acquired using the 488 nm and 647 nm lasers. For dual-colour imaging of ScaC-AF647 and TMR-dynein, we used a DV2 beam splitter (Photometrics), which projected each channel onto a different half of the image. Three replicates were performed for each complex combination. For image analysis, kymographs were generated from tiff movie stacks using Fiji ImageJ software (version 2.8.0). For the quantification of processive events per microtubule µm per minute, processive runs were selected if they moved at least 525nm[15]. Velocity was calculated using a pixel size of 105 nm and a frame rate of 236 ms/frame. The fraction of complexes containing both TMR-dynein and ScaC-AF647 was calculated as the number of dual colour events divided by the total number of TMR-dynein events and by the labelling efficiency of ScaC-AF647 (which was determined to be ~90.6%).

## Immunoblotting analysis and antibodies

For western blot analysis, membranes were blocked with 5% milk in TBST buffer (20 mM Tris, 150 mM NaCl, 0.1% (w/v) Tween 20) for 45 min at room temperature. Primary antibody incubation was carried out in 5% milk in TBST for either 2 h at room temperature or overnight at 4 °C. Secondary antibody incubation was carried out in 5% milk in TBST for 1 h at room temperature. Chemiluminescent detection was performed using ECL western blot detection reagent (Cytiva). Primary antibodies used were: rabbit polyclonal anti-BICD1 (Atlas Antibodies HPA041309, 1:1000 dilution), rabbit polyclonal anti-BICD2 (Atlas Antibodies HPA023013, 1:1000 dilution), mouse monoclonal anti-GAPDH (Abcam, ab8245, 1:1000 dilution), mouse monoclonal anti-β-tubulin (Sigma, T4026). Secondary antibodies used were: goat anti-rabbit IgG HRP (Abcam, ab6721, 1:2000 dilution) and goat anti-mouse IgG HRP (Dako, P0447, 1:1500 dilution).

## AlphaFold prediction

Structure predictions were performed using Alphafold2 or Alphafold2-Multimer through a local installation of ColabFold. The structures of ScaC (UniProt I6Z768) residues 33-223 and BICD2 (UniProt Q921C5) residues 711-800 were predicted by running ColabFold 1.3.0 on two copies of these segments. The interactions between BICD2 (residues 650-800) and Rab6 (UniProt P20340) or RanBP2 (UniProt P49792) residues 2148-2240 were predicted by running ColabFold 1.3.0 on two copies of BICD2 and one copy of either Rab6 or RanBP2.

## Statistical analysis

Statistical analysis was performed in R. Details on the specific statistical test used for each experiment can be found in the respective figure legend. Plots were made using R (RStudio version 4.3.3).

## Reporting summary

Further information on research design is available in the Nature Portfolio Reporting Summary linked to this article.

## Data availability

AlphaFold structure models have been deposited on ModelArchive (modelarchive.org) and are available with the accession codes ma-0p39h (BICD2 residues 711-800), ma-amwrh (ScaC residues 33-223), ma-2nlcr (BICD2 residues 650-800 in complex with RAB6A) and ma-hss3s (BICD2 residues 650-800 in complex with RANBP2 residues 2148-2240). The mass spectrometry proteomics data have been deposited to the ProteomeXchange Consortium via the PRIDE partner repository[63] (http://www.ebi.ac.uk/pride) with the dataset identifier PXD064516. Source data are provided with this paper. In addition, all light microscopy data, proteomics data, western blots, SDS-PAGE gels and statistical analyses are available at https://doi.org/10.5281/zenodo.14888115. Source data are provided with this paper.

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

## Acknowledgements

We thank C.M. Johnson for assistance with SEC-MALS and CD experiments. We thank F. Begum and C. Franco for providing support with mass spectrometry experiments and data analysis. We thank M. Laub for assistance with Python image analysis. We thank S. Bullock and Y. Andres Jeske for help with the HeLa Flp-In T-REX FRT-TO system. We thank K. Singh and C.K. Lau for help with protein purification. We thank F. Randow for helpful discussions. This work was supported by a Boehringer Ingelheim Fonds PhD Fellowship to G.M., a Wellcome Investigator Award to A.P.C. (210711/Z/18/Z and 227434/Z/23/Z), a Wellcome Trust Senior Research Fellowship to J.S. (224277/Z/21/Z) and Medical Research Council funding to A.P.C. (MC_UP_A025_1011).

## Author contributions

G.M. performed cloning, protein purification, biochemical assays, single molecule assays, cellular experiments and data analysis. K.S., P.C. and C.K. performed infection experiments and data analysis. J.B. and K.S. performed image analysis. S.A. performed live imaging of *O. tsutsugamushi*. T.E.M., H.K. and G.M. performed cross-linking mass spectrometry. J.S., A.P.C. and G.M. conceived the project. G.M., J.S. and A.P.C. wrote the manuscript with input from all authors.

## Competing interests

The authors declare no competing interests.
