## [Transparent Peer Review file · Nature Communications]

The intracellular bacterium *Orientia tsutsugamushi* uses the autotransporter ScaC to activate BICD adaptors for dynein-based motility.

Corresponding Author: Dr Andrew Carter

Version 0:

Reviewer comments:

Reviewer #1

(Remarks to the Author)

This manuscript provides important novel insight into how the intracellular bacterial pathogen *Orientia tsutsugamushi* interacts with the host microtubular cytoskeleton via dynein and its adaptor BICD2. I started to read this work with “some suspicion” as I have not been fully convinced by the previous work on dynein-subversion by *Orientia* (published in *I and I* in 2002). However, the great quality of the executed work here and the very careful analysis have completely changed my mind. I find that this manuscript is simply superb and it is of very high value to the scientific community. The ScaC-BICD2-dynein axis is very compelling and a very exciting discovery. I want to congratulate the authors on this fantastic work, and I only have minor comments (see below) that could potentially make this work a bit more complete.

Specific comments:

I am fully convinced by the data presented by the authors on the hijacking of dynein via BICD2 through ScaC from the bacterium. The authors follow in their thoughts the previous publication on relocalization of the bacteria to the perinuclear region (*I and I* paper from 2002). For me, the quantification shown in Fig2B does not really seem to represent the figures in 2A- the phenotype seems to be much stronger in the images compared to the quantification. Could it be that something went wrong? Could the authors alternatively label the MTOC and perform quantifications on MTOC-bacteria distances? Is there another way to quantify?

Another great addition (perhaps beyond the scope of this manuscript) could be the use of holotomography imaging (label-free; commercialized by Nanolive or Tomocube for example). This approach is particularly powerful for researchers working with bacteria that cannot be genetically modified. It's a simple suggestion, I understand if the authors cannot carry this out, but it would have the potential to give a great addition to the work. It would allow for example the tracing of individual bacteria over time, and the authors could learn whether the bacteria move on individual microtubular tracks, or whether they “jump” between tracks.

The “tour-de-force” figure 3 is great!

Additional points:

There is one report on the role of dynein in *Orientia* intracellular movements (cited also by the authors). I would have appreciated a bit more critical review of this manuscript, for example the old work uses methanol fixations and co-localizations are not really quantified well (or at all).

The group of bacteria hijacking microtubules is growing, there is recent work on *Shigella* exploiting the dynein-microtubular axis for vacuolar uncoating. This work may also be cited and discussed.

The capacity of ScaC to modulate BICD2 to enable an open conformation for dynein-dynactin assembly enables controlled interference with cellular processes requiring these proteins. It would be great to have one paragraph in the discussion how ScaC could be used to mechanistically study such processes.

Reviewer #2

(Remarks to the Author)

Previous work has demonstrated that dynein plays a role in establishing *O. tsutsugamushi* perinuclear localization within infected cells. However, the factors that are involved in bacteria-dynein coordination was unclear. In this work, Manigrasso et al. showed that *O. tsutsugamushi* cell surface protein ScaC directly interacts with dynein activating adaptor BICD2 using multiple in vitro pull-down assays. The ScaC-BICD2 interaction interface was mapped using truncated mutants and cross-linking MS. The authors further demonstrated that ScaC-BICD2 interaction facilitates the activation of dynein by the previously established motility assay, implicating the potential mechanism of *O. tsutsugamushi* movement along microtubules. The interesting observation of direct binding of a bacterial protein to the dynein activating adaptor represent a new paradigm of pathogenic manipulation of the dynein motor in addition to the reported roles in translocating bacteria-containing vacuoles within infected cells. The weakest point of this manuscript is on the limited evidence on whether ScaC-BICD2 interaction drives the bacterial movement along microtubules via dynein.

Although it is shown in the manuscript that the purified ScaC protein can activate dynein in the motility assay, there is no direct evidence that ScaC on the surface of *O. tsutsugamushi* mediates bacterial movement along microtubules. Moreover, the localization of ScaC and BICD1/BICD2 in infected cells has not been shown. The same motility assay using the bacteria as "cargo" may possibly address this issue.

The disruption of perinuclear translocation of *O. tsutsugamushi* under siRNA treatment was mild (Figure 2). One possibility is that the ScaC on the bacterial surface is barely accessible to BICD2. Another possibility is that ScaC-BICD2 interaction only plays a minor role in the bacterial translocation and other unidentified factors play a more significant role. Silencing other dynein activating adaptors besides BICD1 and BICD2 should be considered.

Along this line, the authors discussed the possible differential expression of BICD1 vs BICD2 (or other dynein activating adaptors) in various cell types. Would ScaC interacts with other dynein activating adaptors (e.g. related BICD family members, BICDR1 and BICDR2) in different cell types, esp. in a more physiologically relevant condition?

Minor comments:

- Considering that the dynein activating adaptors coordinate a wide spectrum of cargos to the dynein motor for cargo transport, why is *O. tsutsugamushi* specifically hijacking BICD2? Is BICD2 ubiquitous expressed in various cell types? Is it generally more abundantly expressed among other adaptors? Does it offer some insights into the functions of BICD2 for molecular transport?
- The actin-based motility of *Listeria* or other bacterial species occurs within a small-time frame once the bacteria enter the host cytosol. Whether the perinuclear translocation of *O. tsutsugamushi* helps the bacteria to evade host defenses can be easily tested using immunostaining with known autophagic markers.
- Some typos in the figure legend of Figure 3.

Reviewer #3

(Remarks to the Author)

Manigrasso et al. report that the *Orientia tsutsugamushi* surface-localized autotransporter ScaC binds the dynein adaptor BICD2 and that BICD2 is facilitator of *Orientia* perinuclear localization. Ectopically expressed ScaC is sufficient to bind BICD2 (and BICD1). The BICD2 C-terminus is required for interacting with ScaC, and this interaction is at least partially independent of known binding interfaces of other BICD2 interaction partners. They show that ectopically expressed ScaC activates BICD2 for dynein motility. From their data, almost all of which are derived from studying recombinant/ectopically expressed ScaC, the authors extrapolate that ScaC engages BICD2 for *Orientia* to move along microtubules to the perinuclear region where replication occurs. The ScaC-BICD2 interaction and dynein activation studies provide are solid and begin to support the authors' working model, which very well could be correct. If so, then this would provide novel insight to how the infection process of an understudied pathogen. Unfortunately, the understanding gained thus far is superficial primarily because, aside from the BICD1/2 knockdown experiment, the study lacks experimental data in the context of infection. Much more validation work and controls are needed to justify their interpretation. The authors are capable of rounding out this potentially interesting story.

Major concerns:

1. In the abstract (lines 21-22), introduction (lines 64-55) and throughout, it is stated that *O. tsutsugamushi* uses ScaC to recruit BICD2 to the bacterial surface. This is a presumption that has not been experimentally demonstrated. The authors presume that ScaC is sufficient for bacterial intracellular motility, but this is also not experimentally demonstrated. I appreciate the challenges of working with an obligate, but there are several options at least some of which could (should) be explored:
 - a. Use *Orientia* in a pull-down assay to see if it can precipitate BICD1/2 as Martinez did for the *Rickettsia* receptor Ku70 (PMC16360032).
 - b. Use superresolution or other high resolution fluorescence microscopy to demonstrate close association of ScaC on the *Orientia* surface and BICD1/2 in host cells.
 - c. Immunoelectron microscopy as an alternative to "b".

2. All of the experiments are performed in HeLa epithelial cells, which presents issues. Although HeLa cells have proven effective models to study *O. tsutsugamushi*-host cell interactions in the literature, they alone are not the most relevant model for this project. As suggested in lines 258-260, BICD1 – not BICD2 - has the potential to be a relevant facilitator of *O. tsutsugamushi* microtubule trafficking in endothelial cells, which are a natural host cell type in scrub typhus and more physiologically relevant. In addition, Figure 1 supports that ScaC also interacts with BICD1. The author's argument that BICD1 is relevant to infection in endothelial cells should be tested.

3. I have several concerns about the BICD1/2 knockdown and infection experiment. First and foremost, the authors state that an MOI of 50 was used. Yet, there are barely individual bacteria per cell in the representative images. In looking at their methods, they inoculate with frozen bacteria. Clearly, viability is affected. What else in terms of infection or the very interaction being studied is compromised by using frozen bacteria? Also, only one postinfection time point is examined. And, if *Orientia* getting to its normal perinuclear location is altered, what is the effect on replication, overall infection fitness, and completion of the infection cycle?

4. BICD2 was identified as an interacting partner of ScaC in bulk IP-proteomics screen and via protein-protein pull-down assays, which rely on tagged proteins and non-native conditions. It should be proven that the ScaC-BICD1/2 interaction occurs during infection using at least one endogenous binding partner.

5. Based on previous literature, nocodazole disrupts *O. tsutsugamushi* MTOC localization and as such was used as a positive control for mitochondria relocation experiments in Supplementary Fig 1. Because this positive control is available, it would be prudent to quantify the impact nocodazole has on *O. tsutsugamushi* localization compared to BICD1 and BICD2 RNAi knockdown in Figure 2. The RNAi technique does not seem to completely abolish *O. tsutsugamushi* movement like seen in Kim et al. PMC97908, suggesting other mechanisms are involved in facilitating *O. tsutsugamushi* perinuclear localization.

6. A number of additional experiments lack relevant controls and/or validation. (i) Other GFP-Sca fusions rather than GFP alone should be included as negative controls for GFP-ScaC mitochondrial localization and protein-protein interaction studies. (ii) BICD1 and BICD2 expression in normal HeLa cells should be included alongside the HeLa Flp-In cells. (iii) What is BICD1/2 expression infection during *Orientia* infection?

7. To my knowledge, ScaC dimerization in *O. tsutsugamushi* has not been assessed previously. Does ScaC dimerization happen naturally? Is the dimerization required for binding to BICD2? Is BICD2 dimerization required for interaction with ScaC?

8. Lines 191-193 and Figure 3 suggests that the BICD2 binding region for ScaC is novel. The portion of BICD2 required for ScaC to bind is 89 amino acids long with two different regions dismissed as potential binding sites in Figure 3. Is there a region of the protein within BICD2711-800 that points to being this novel binding site based on the characteristics of the proteins and/or knowledge potentially present within the literature? From my understanding, RanBP2 and Rab6 are not the only interaction partners of BICD2. Can you really say that ScaC-BICD2 interaction domain is a novel domain, if the portion of BICD2 required for ScaC binding wasn't identified? To this end, the ScaC binding domain of BICD2 should be flushed out.

9. What is the impact of knocking down BICD1 and BICD2 on microtubule localization and structure? Does *O. tsutsugamushi* and/or still localize with and/or activate microtubules without BICD2?

Minor

1. Line 34-37, *Chlamydia trachomatis* does not move along microtubules. Rather, the chlamydial inclusion traffics on microtubules. Also, chlamydiae are not encased in endocytic membranes per se. It is better to refer to it as the inclusion membrane.

2. What is the level of conservation between ScaC and other Sca proteins? This is important to consider if ScaC is the unique Sca to bind BICD1/2 and why other Sca proteins should be included as controls. Such information could help inform experiments to pin down the region of ScaC required for binding BICD1/2.

3. The circular dichroism data should be better explained in the results and/or legend to be digestible by a broader audience.

4. Lines 186-188 are confusing. Because BICDR1 has a Rab6 binding site is the very reason it cannot interact with ScaC???

Reviewer #4

(Remarks to the Author)

In this manuscript, Manigrasso et al. demonstrated that *O. tsutsugamushi* recruits the activating adaptors BICD1 and BICD2 for dynein-based retrograde trafficking. Through SEC and single molecule assays, they showed that the outer membrane passenger domain of ScaC from *O. tsutsugamushi* interacts with the CC3 domain of BICD2, releasing its auto-inhibition and

making its N-terminus available for interaction with dynein and dynactin. They also demonstrated that BICD2 is necessary for the nuclear migration of *O. tsutsugamushi*.

The in vitro data are of high quality, and the system and findings hold significant potential for the field. However, there are a few minor caveats that are not adequately addressed, and some important controls related to the key experiments need to be conducted.

Since the ScaB autotransporter is essential for the invasion of mammalian cells by *Orientia tsutsugamushi*, the authors need to provide a more robust explanation for why they did not study the GFP-ScaB co-IP/MS in HeLa cells.

Cell lines such as monocytes/macrophages and dendritic cells are more biologically relevant hosts for *O. tsutsugamushi*. To avoid overinterpreting the results, the overexpression of ScaA/C/D/E and its interaction with BICD1/2 should be reported in at least one of these cell lines.

Furthermore, in supporting figure 4, the authors have shown that BICD1 has almost zero expression in HeLa cells (Fig S4 K) and they showed the gene expression level of BICD2, measured by qPCR, are ~57-fold higher than those of BICD1 in the cell model. Therefore, its knockdown in HeLa cells cannot provide conclusive results regarding ScaC trafficking. At least one more cell line with higher BICD1 expression needs to be included in this study (such as lymphatic endothelial cells that they mentioned in the discussion part).

The results indicate that GFP-ScaCMTS expression mimicked the effect of GFP-BICD2NMTS. Therefore, the authors need to provide data showing the localization of over-expressed GFP-ScaC itself (without mitochondrial attachment) within the cell. Can they observe any perinuclear accumulation? Additionally, why do mitochondria bound to ScaC accumulate at 1-3 single perinuclear points? Shouldn't they be distributed around the nucleus?

Does this substantial accumulation of GFP-ScaCMTS (mitochondria) at the nucleus impact the distribution of other organelles within the cells? The authors should provide evidence that cells remain healthy upon overexpression of ScaC.

The impact of BicD2 knockdown on nuclear localization (as shown in Figure 2) appears to be relatively minor. The authors should also present the results of the BicD2 knockout experiment.

Furthermore, it is essential to report a "direct" infectivity phenotype following BicD2 knockdown/out, such as deficiencies in replication or the number of daughter bacteria formed inside the cell.

Btw, authors need to clarify what TSA56 represents as a phenotype of bacterial infection? There is no information provided on this in the results section.

Although cross-linking experiments demonstrated that residues 119-136 of ScaC interact with the central region of BicD2 CC3, the researchers opted to study the interaction between truncated ScaC (residues 50-159) and BicD2 in more detail. The researchers need to report the cross-linking results for residues 119-136 of ScaC with BicD2-CC3.

In Figure 4A, it appears that the authors used the incorrect colors for the lines in the chromatogram. The red line should correspond to "BICD2N + BICD2C + ScaC," while the black line should represent "BICD2N + BICD2C" only.

"As purified viral capsids were found to interact with the BICD2 CC3" → Carnes et al (ref 47) weren't able to show the binding of CC3 to the assembled capsid. Line 267 needs to be corrected accordingly.

Version 1:

Reviewer comments:

Reviewer #2

(Remarks to the Author)

The authors have revised the manuscript with additional experiments, which has sufficiently addressed my concerns to the initial version of the manuscript. The manuscript is well-written. The new experimental data with the BICD adaptors knock-out cells supported the role of dynein (via the interaction with BICD1 and BICD2) on perinuclear clustering of the bacteria.

The revised manuscript offers new insights on how bacterial pathogens exploit the host molecular motors for establishing their intracellular niche. The reported observations will stimulate further discovery of the field.

Reviewer #3

(Remarks to the Author)

I appreciate the authors' response to my and the other reviewers' concerns. The additional data, particularly the use of BICD1 and BICD2 KO lines, nocodazole, and dominant negative ScaC in the context of infection when combined with the strong biochemical data make for a compelling story that the ScaC-BICD1/2 interaction contributes to *Orientia mobilization* along microtubules. I have only minor comments at this point.

1. I am still unconvinced by the response that freezing host cell-free *Orientia* does not impact its viability/fitness. First, the

Giengkam 2015 paper does not directly compare the growth of *Orientia* in cells after being frozen in various buffers to *Orientia* that had not been frozen (Fig 7C). Second, in looking at the IF images in the response to critique 2.1, at 2 dpi there are barely any bacteria in the cell. In fact, I would argue that there are 1-2 log-fold differences in the number of bacteria in these cells vs if the inocula had come from actively growing *Orientia* infected cells. If your MOI was 50 and you only see 4-5 intracellular bacteria at 2 dpi, then the viability of your inoculum was compromised.

That said, the *Orientia* load in WT cells in Figure 4E is much higher than in the aforementioned data. Because 4E data is what was included in the manuscript and it supports the hypothetical model, I won't belabor this point further. At some point, however, I suggest that you directly compare the growth of equivalent numbers of live *Orientia* recovered from infected cells vs frozen host cell-free *Orientia*. Adding so much dead frozen and thawed bacteria is a concern and could ultimately yield misleading results depending on what you are examining, especially if you ever look at host immune responses.

2. Lines 204-206 (Fig. 3A) – After reading and looking over Figure 3A several times, I do not see how the chromatogram supports this statement. Please revise to provide more detail or clue me in.

3. Line 52 – I'm not sure if phagosomal is the best adjective here given *Orientia* endocytic entry. Perhaps endosome. Or if you want to go more generic to toe the line between endosome and phagosome, then perhaps vacuolar.

4. Line 129 – I suggest moving (Fig. 1H-I) to the end of the sentence on line 132.

5. Supplementary figure 2L – italicize GAPDH since you are measuring gene expression.

6. Lines 213, 218, 230 – you pick and choose when you want to list values. I'm not sure this is necessary especially for the 3-fold difference (lines 213-124, Figure 3B-C) as the fold difference is clear via the bar graph. Ultimately, this is up to you, and is a suggestion.

7. 249-250 – If this manuscript moves forward, the Nat Comm copy editors will likely weigh in on this, but I do not think you can refer to data and not show it. I suggest adding to Supplementary figure 6 as it does easily set up why you went for the CRISPR knock outs of BICD1/2.

8. Lines 280-283 – Similarly, I do not think you can reference a BioRxiv manuscript as it has not been peer reviewed. I suggest striking this information and reference unless it gets published in time.

Reviewer #4

(Remarks to the Author)

I appreciate the authors for addressing all my concerns. They have conducted the related experiments excellently, and I strongly recommend the current version of this manuscript for publication in Nature Communications.

REVIEWER COMMENTS

Reviewer #1 (Remarks to the Author):

This manuscript provides important novel insight into how the intracellular bacterial pathogen *Orientia tsutsugamushi* interacts with the host microtubular cytoskeleton via dynein and its adaptor BICD2. I started to read this work with “some suspicion” as I have not been fully convinced by the previous work on dynein-subversion by *Orientia* (published in *J Cell Biol* 2002). However, the great quality of the executed work here and the very careful analysis have completely changed my mind. I find that this manuscript is simply superb and it is of very high value to the scientific community. The ScaC-BICD2-dynein axis is very compelling and a very exciting discovery. I want to congratulate the authors on this fantastic work, and I only have minor comments (see below) that could potentially make this work a bit more complete.

Specific comments:

I am fully convinced by the data presented by the authors on the hijacking of dynein via BICD2 through ScaC from the bacterium. The authors follow in their thoughts the previous publication on relocalization of the bacteria to the perinuclear region (*J Cell Biol* 2002). For me, the quantification shown in Fig2B does not really seem to represent the figures in 2A- the phenotype seems to be much stronger in the images compared to the quantification. Could it be that something went wrong? Could the authors alternatively label the MTOC and perform quantifications on MTOC-bacteria distances? Is there another way to quantify?

The disruption of bacterial clustering at the perinucleus was relatively weak with siRNA knockdowns. We hypothesised that this was due to the remaining BICD1 or BICD2 being sufficient for transport. We therefore over expressed ScaC to act as a dominant negative and in parallel generated HeLa lines in which BICD1 and BICD2 were knocked out either individually or in combination. Both approaches had a much larger effect on clustering, equivalent that observed upon nocodazole treatment (which causes depolymerization of microtubules). We have therefore removed the siRNA data from the manuscript and replaced the former Figure 2 with a new Figure 4 showing our new data.

1.1) Another great addition (perhaps beyond the scope of this manuscript) could be the use of holotomography imaging (label-free; commercialized by Nanolive or Tomocube for example). This approach is particularly powerful for researchers working with bacteria that cannot be genetically modified. It's a simple suggestion, I understand if the authors cannot carry this out, but it would have the potential to give a great addition to the work. It would allow for example the tracing of individual bacteria over time, and the authors could learn whether the bacteria move on individual microtubular tracks, or whether they “jump” between tracks.

This is a great suggestion, but unfortunately, we tried this several years ago and it doesn't work for live imaging of *Orientia* (or *Rickettsia*). We found that the signal of the bacteria was not sufficiently different from the background for us to be able to resolve them. We would also need one of these microscopes within a CL3 facility which would be problematic. We have now included some live cell imaging data in

which bacteria were fluorescently labelled using a fluorescent probe (refer to Supplementary Figure 6 A-B).

The "tour-de-force" figure 3 is great!

Additional points:

1.2) There is one report on the role of dynein in *Orientia* intracellular movements (cited also by the authors). I would have appreciated a bit more critical review of this manuscript, for example the old work uses methanol fixations and co-localizations are not really quantified well (or at all).

Whilst we agree that this previous paper lacked quantification and controls, we feel that since our paper confirms its main findings, there is not a need to elaborate on the weaknesses of that work.

1.3) The group of bacteria hijacking microtubules is growing, there is recent work on *Shigella* exploiting the dynein-microtubular axis for vacuolar uncoating. This work may also be cited and discussed.

Thank you for pointing this out. We have now included a mention of this in the first paragraph of the discussion "It has also recently been shown that a different cytosolic bacterium, *Shigella flexneri*, harnesses force from dynein through activating adaptors to drive uncoating of the bacteria-containing vacuole (PMID 38316786)."

1.4) The capacity of ScaC to modulate BICD2 to enable an open conformation for dynein-dynactin assembly enables controlled interference with cellular processes requiring these proteins. It would be great to have one paragraph in the discussion how ScaC could be used to mechanistically study such processes.

This is a great suggestion! We have added it to the end of the discussion.

Reviewer #2 (Remarks to the Author):

Previous work has demonstrated that dynein plays a role in establishing *O. tsutsugamushi* perinuclear localization within infected cells. However, the factors that are involved in bacteria-dynein coordination was unclear. In this work, Manigrasso et al. showed that *O. tsutsugamushi* cell surface protein ScaC directly interacts with dynein activating adaptor BICD2 using multiple in vitro pull-down assays. The ScaC-BICD2 interaction interface was mapped using truncated mutants and cross-linking MS. The authors further demonstrated that ScaC-BICD2 interaction facilitates the activation of dynein by the previously established motility assay, implicating the potential mechanism of *O. tsutsugamushi* movement along microtubules. The interesting observation of direct binding of a bacterial protein to the dynein activating adaptor represent a new paradigm of pathogenic manipulation of the dynein motor in addition to the reported roles in translocating bacteria-containing vacuoles within infected cells. The weakest point of this manuscript is on the limited evidence on

whether ScaC-BICD2 interaction drives the bacterial movement along microtubules via dynein.

To provide better evidence that the ScaC-BICD2 interaction drives bacterial movement we include two new experiments which we believe strengthen the manuscript. As described in our response to reviewer 1 we over-expressed ScaC (to act as a dominant negative) and made Hela lines in which BICD1, BICD2 or both adaptors were knocked out. Both approaches had a much larger effect on clustering than previous siRNA approaches. The inhibition of perinuclear clustering was equivalent to nocodazole treatment to remove microtubules (new Figure 4) suggesting that ScaC is the key bacterial component for this pathway and that BICD1/2 are sufficient for transport.

2.1) Although it is shown in the manuscript that the purified ScaC protein can activate dynein in the motility assay, there is no direct evidence that ScaC on the surface of *O. tsutsugamushi* mediates bacterial movement along microtubules. Moreover, the localization of ScaC and BICD1/BICD2 in infected cells has not been shown. The same motility assay using the bacteria as “cargo” may possibly address this issue.

In addition to our new data in Figure 4 we attempted to visualize the recruitment of BICD2 to the bacterial surface during infection through immunofluorescence staining. However, whilst BICD2 labelling was observed in the vicinity of the bacteria, we cannot definitely demonstrate that this is a direct interaction (see Figure below). We suspect this is due to the low surface abundance of ScaC.

Supporting this hypothesis, a previous study by the Salje lab (PMC7335160) using dual RNAseq of *O. tsutsugamushi* and endothelial HUVEC cells at 5 dpi reported ScaC expression at 8.92 transcripts per million (tpm), which is 99-fold lower than TSA56 (886.84 tpm), the bacterial surface protein we used as a marker in our immunofluorescence experiments. These data are shown here:

Using bacteria as cargo in motility assays would be ideal, but unfortunately *O. tsutsugamushi* is classified as a Containment Level 3 pathogen and we do not have the microscopes needed for in vitro motility assays in our containment facility.

2.2) The disruption of perinuclear translocation of *O. tsutsugamushi* under siRNA treatment was mild (Figure 2). One possibility is that the ScaC on the bacterial surface is barely accessible to BICD2. Another possibility is that ScaC-BICD2 interaction only plays a minor role in the bacterial translocation and other unidentified factors play a more significant role. Silencing other dynein activating adaptors besides BICD1 and BICD2 should be considered.

As the reviewer suggested, the partial inhibition of perinuclear transport of *O. tsutsugamushi* following siRNA treatment could be attributed to the presence of other, BICD1/2-independent pathways that contribute to bacterial transport. An alternative explanation is that the residual levels of BICD1 and BICD2 in the siRNA-treated cells are sufficient to drive the movement of the bacteria. We have now generated single knockout BICD1^{-/-} and BICD2^{-/-} HeLa cell lines, as well as a double BICD1^{-/-} / BICD2^{-/-} line (refer to Supplementary Fig. S6), and showed that the perinuclear translocation of *O. tsutsugamushi* is completely disrupted in BICD1^{-/-} / BICD2^{-/-} cells (refer to Fig. 4C-D). This result strongly suggests that there are no other activating adaptors involved in this process. We now state that BICD1 and BICD2 are sufficient for *O. tsutsugamushi* accumulation next to the nucleus (Lines 256-257)

2.3) Along this line, the authors discussed the possible differential expression of BICD1 vs BICD2 (or other dynein activating adaptors) in various cell types. Would ScaC interact with other dynein activating adaptors (e.g. related BICD family members, BICDR1 and BICDR2) in different cell types, esp. in a more physiologically relevant condition?

In Supplementary Figure S4D we show that ScaC does not bind BICDR1, indicating that the interaction with BICD1 and BICD2 is very specific and does not extend to other related BICD family members. Further, as mentioned above, our observation that a double BICD1/BICD2 knockout fully inhibits perinuclear clustering of the bacteria suggests that BICD1 and BICD2 are together both necessary and sufficient for transport.

Minor comments:

2.4) Considering that the dynein activating adaptors coordinate a wide spectrum of cargos to the dynein motor for cargo transport, why is *O. tsutsugamushi* specifically hijacking BICD2? Is BICD2 ubiquitously expressed in various cell types? Is it generally more abundantly expressed among other adaptors? Does it offer some insights into the functions of BICD2 for molecular transport?

This is an intriguing question, though we can only provide a speculative answer at this stage. BICD2 is ubiquitously and abundantly expressed in many cell types, which could be a desirable feature for a pathogen with a widespread cell tropism like *O. tsutsugamushi*. Our immunofluorescence data (see Figure 1F) shows that BICD2 localizes to perinuclear vesicles (likely Rab6-positive secretory vesicles) while also maintaining a cytosolic pool, which makes it readily accessible to the bacterium after phagosomal escape. Despite this observation, we cannot say for sure why *O. tsutsugamushi* specifically hijacks BICD2 (and its paralogue BICD1). While some dynein adaptors such as BICD1, BICDR1 and JIP3, are largely brain-specific, others like HOOK3 are also widely expressed across various cell types.

2.5) The actin-based motility of *Listeria* or other bacterial species occurs within a small-time frame once the bacteria enter the host cytosol. Whether the perinuclear translocation of *O. tsutsugamushi* helps the bacteria to evade host defenses can be easily tested using immunostaining with known autophagic markers.

We have carried out extensive experiments using autophagy markers but have not been able to clearly see co-labelling with *Orientia* under the conditions we tested. Nonetheless, we remain interested in exploring whether *O. tsutsugamushi* is targeted by autophagy and if dynein-driven movement helps the bacterium escape autophagic degradation. Investigating the interplay between *O. tsutsugamushi*, microtubule-based transport and autophagy will be a focus of follow-on studies.

2.6) Some typos in the figure legend of Figure 3.

We thank the reviewer for spotting these mistakes, which we have now corrected.

Reviewer #3 (Remarks to the Author):

Manigrasso et al. report that the *Orientia tsutsugamushi* surface-localized autotransporter ScaC binds the dynein adaptor BICD2 and that BICD2 is facilitator of *Orientia* perinuclear localization. Ectopically expressed ScaC is sufficient to bind BICD2 (and BICD1). The BICD2 C-terminus is required for interacting with ScaC, and this interaction is at least partially independent of known binding interfaces of other BICD2 interaction partners. They show that ectopically expressed ScaC activates BICD2 for dynein motility. From their data, almost all of which are derived from studying recombinant/ectopically expressed ScaC, the authors extrapolate that ScaC engages BICD2 for *Orientia* to move along microtubules to the perinuclear region where replication occurs. The ScaC-BICD2 interaction and dynein activation studies provide solid and begin to support the authors' working model, which very well could be correct. If so, then this would provide novel insight to how the infection process of an understudied pathogen. Unfortunately, the understanding

gained thus far is superficial primarily because, aside from the BICD1/2 knockdown experiment, the study lacks experimental data in the context of infection. Much more validation work and controls are needed to justify their interpretation. The authors are capable of rounding out this potentially interesting story.

Major concerns:

3.1) In the abstract (lines 21-22), introduction (lines 64-55) and throughout, it is stated that *O. tsutsugamushi* uses ScaC to recruit BICD2 to the bacterial surface. This is a presumption that has not been experimentally demonstrated. The authors presume that ScaC is sufficient for bacterial intracellular motility, but this is also not experimentally demonstrated. I appreciate the challenges of working with an obligate, but there are several options at least some of which could (should) be explored:

- a. Use *Orientia* in a pull-down assay to see if it can precipitate BICD1/2 as Martinez did for the *Rickettsia* receptor Ku70 (PMC16360032).
- b. Use superresolution or other high resolution fluorescence microscopy to demonstrate close association of ScaC on the *Orientia* surface and BICD1/2 in host cells.
- c. Immunoelectron microscopy as an alternative to “b”.

We appreciate this concern, and we believe that the new data using the ScaC overexpression cell line and a BICD1/BICD2 knockout cell line now demonstrate a direct role for ScaC and BICD1/2 in bacterial motility.

We carried out several attempts at the proposed pulldown experiment but unfortunately these were not successful due to high background in the control samples. A description of our attempts at the fluorescence microscopy experiments is included in our response to reviewer comment 2.1.

In both cases we have come to think that the experiments were not feasible due to the very low detectable expression levels of ScaC during cellular infection. We know that the ScaC expression levels are low based on previous proteomics and RNAseq data from our lab (Mika-Gospodorz et al 2020, Giengkam et al, 2023), We also know that ScaC levels are finely tuned throughout the bacterial infection cycle (Giengkam et al 2023 and more recent unpublished time course data from our lab). This underscores the challenge of carrying out the proposed experiments. Given the known low levels of ScaC we felt that immunoelectron microscopy would not be useful as it would not be possible to distinguish signal from background.

3.2) All of the experiments are performed in HeLa epithelial cells, which presents issues. Although HeLa cells have proven effective models to study *O. tsutsugamushi* host cell interactions in the literature, they alone are not the most relevant model for this project. As suggested in lines 258-260, BICD1 – not BICD2 - has the potential to be a relevant facilitator of *O. tsutsugamushi* microtubule trafficking in endothelial cells, which are a natural host cell type in scrub typhus and more physiologically relevant. In addition, Figure 1 supports that ScaC also interacts with BICD1. The author’s argument that BICD1 is relevant to infection in endothelial cells should be tested.

In our revised manuscript we have further explored the relative importance of both BICD1 and BICD2 further. Importantly, as also described in response to reviewer 1&2, we generated knockout cells (in HeLa) for both BICD1 and BICD2, either individually or in combination. Our revised Fig. 4 shows that removal of both genes is required to completely abolish bacterial perinuclear localisation. Therefore, whilst BicD2 is more highly expressed, and more relevant for motility in HeLa cells, the residual BicD1 is sufficient to confer significant bacterial trafficking to the perinuclear region. Incidentally, this observation supports our finding (described in the previous point) that ScaC levels are low and difficult to detect during infection. Our model is that only a small number of ScaC and BICD1/2 molecules are required to drive bacterial motility.

O. tsutsugamushi is also transported to the perinuclear region in an endothelial cell line (HUVEC). We checked the relative levels of BICD1 and BICD2 in HUVECs and found that BICD2 is still more abundant than BICD1 (by approximately 4-fold). We have therefore removed our argument that BICD1 may be relevant to infection in endothelial cells.

3.3) I have several concerns about the BICD1/2 knockdown and infection experiment. First and foremost, the authors state that an MOI of 50 was used. Yet, there are barely individual bacteria per cell in the representative images. In looking at their methods, they inoculate with frozen bacteria. Clearly, viability is affected. What else in terms of infection or the very interaction being studied is compromised by using frozen bacteria? Also, only one postinfection time point is examined. And, if *Orientia* getting to its normal perinuclear location is altered, what is the effect on replication, overall infection fitness, and completion of the infection cycle?

We have now removed this data and replaced it with equivalent experiments carried out in BICD1/2 knockout cell lines (see Figure 4).

We have previously measured the effect of freezing on bacterial viability and show that it does not significantly affect growth (Giengkam et al 2015, PMC7335160). The use of frozen aliquots increases reproducibility, since all experiments can be carried out using bacteria that are prepared in the same way.

In preparing for this manuscript, we measured bacterial position at different times after infection and found that the one presented (i.e. 2 dpi) gave the clearest result in terms of allowing sufficient time for bacteria to reach the perinucleus (in untreated cells) whilst not undergoing so much replication that it becomes difficult to count and quantify the bacteria.

The question about the role of transport in the bacterial fitness and lifecycle is interesting. The ability to significantly disrupt perinuclear clustering (by ScaC overexpression or BICD1/2 knockout) now allows us to include growth curve data (Supplementary Figure 6C). Intriguingly, blocking perinuclear transport has no effect on growth in HeLa cells. However, as we outline in a new discussion section, we suspect the effect may be clearer *in vivo*. As we discuss in the revised manuscript (Lines 275-292) this hypothesis agrees with other data from our lab.

3.4) BICD2 was identified as an interacting partner of ScaC in bulk IP-proteomics screen and via protein-protein pull-down assays, which rely on tagged proteins and non-native conditions. It should be proven that the ScaC-BICD1/2 interaction occurs during infection using at least one endogenous binding partner.

In our initial co-IP experiment (Figure 1B), ScaC is GFP-tagged but BICD1 and BICD2 are endogenous.

We attempted to detect the ScaC-BICD2 interaction in the context of the whole bacterium using pull-down assays, but these efforts were unsuccessful due to technical challenges. Initially, we tried isolating bacteria from infected cells by centrifugation, but this approach failed to effectively separate the bacteria from the unbound supernatant even after several washes. We then tried mixing purified bacteria with Strep-tagged BICD2 protein and isolating the complex using magnetic anti-Strep beads. However, the bacteria bound to the beads non-specifically, thus affecting the interpretation of the results.

Despite not being able to demonstrate a direct interaction between ScaC and BICD2 on the bacterial surface, we believe the new data presented in the manuscript (Figure 4) strongly supports our model as discussed above.

3.5) Based on previous literature, nocodazole disrupts *O. tsutsugamushi* MTOC localization and as such was used as a positive control for mitochondria relocation experiments in Supplementary Fig 1. Because this positive control is available, it would be prudent to quantify the impact nocodazole has on *O. tsutsugamushi* localization compared to BICD1 and BICD2 RNAi knockdown in Figure 2. The RNAi technique does not seem to completely abolish *O. tsutsugamushi* movement like seen in Kim et al. PMC97908, suggesting other mechanisms are involved in facilitating *O. tsutsugamushi* perinuclear localization.

This is a good suggestion. We now include data on nocodazole treatment of HeLa ATCC cells and demonstrate that it strongly inhibits the perinuclear translocation of *O. tsutsugamushi* (Figure 4), consistent with the findings of Kim *et al.* As described, in response to the other reviewers, we have replaced the siRNA knockdowns with ScaC-GFP overexpression and BICD1/BICD2 knockout cell lines. These treatments show strong inhibition of perinuclear translocation that is similar to the effects of nocodazole.

3.6) A number of additional experiments lack relevant controls and/or validation. (i) Other GFP-Sca fusions rather than GFP alone should be included as negative controls for GFP-ScaC mitochondrial localization and protein-protein interaction studies. (ii) BICD1 and BICD2 expression in normal HeLa cells should be included alongside the HeLa Flp-In cells. (iii) What is BICD1/2 expression infection during *Orientia* infection.

- (i) We purified the passenger domain of ScaE and showed that it does not interact with BICD2 *in vitro* using size exclusion chromatography (see

Supplementary Figure S3E). Unfortunately, we were unable to do the same experiments with the passenger domains of ScaD and ScaA as our attempts to express them in *E. coli* were not successful. We apologise that we were not able to include these fusions in the mitochondrial relocation assay due to time constraints, but note that there was no evidence for their interaction with any adaptors in our mass spectrometry data (Supplementary Figure 1)

(ii) We conducted qPCR analysis of *BICD1* and *BICD2* in HeLa ATCC cells and found that, like in HeLa Flp-In cells, *BICD2* expression was significantly higher compared to *BICD1* (~32-fold difference) (see Supplementary Figure S6D). This explains why *BICD1* knockout alone does not affect the perinuclear translocation of *O. tsutsugamushi* (shown in Figure 4).

(iii) As requested we assessed the protein levels of *BICD1* and *BICD2* at 48 hpi by western blot and found that these were comparable to those in uninfected cells (see Figure below). This result suggests that *O. tsutsugamushi* does not directly modulate the expression of *BICD1* and *BICD2*.

3.7) To my knowledge, ScaC dimerization in *O. tsutsugamushi* has not been assessed previously. Does ScaC dimerization happen naturally? Is the dimerization required for binding to *BICD2*? Is *BICD2* dimerization required for interaction with ScaC?

ScaC dimerization has indeed not been studied previously. Our finding that the passenger domain of ScaC forms a dimer *in vitro* is supported by AlphaFold structure predictions (which suggest that the passenger domain of ScaC dimerizes even in the context of the full-length protein) as well as other structure prediction tools. For instance, the coiled-coil prediction tool COILS predicts with high confidence that the passenger domain of ScaC contains a coiled-coil region. How ScaC dimerization occurs on the bacteria membrane is an intriguing question that we will explore in future work.

BICD2 is a well-characterized coiled-coil protein which predominantly exists as a dimer.

3.8) Lines 191-193 and Figure 3 suggests that the BICD2 binding region for ScaC is novel. The portion of BICD2 required for ScaC to bind is 89 amino acids long with two different regions dismissed as potential binding sites in Figure 3. Is there a region of the protein within BICD2711-800 that points to being this novel binding site based on the characteristics of the proteins and/or knowledge potentially present within the literature? From my understanding, RanBP2 and Rab6 are not the only interaction partners of BICD2. Can you really say that ScaC-BICD2 interaction domain is a novel domain, if the portion of BICD2 required for ScaC binding wasn't identified? To this end, the ScaC binding domain of BICD2 should be flushed out.

We agree with the reviewer that our claim that ScaC interacts with BICD2 through a novel binding interface should be toned down, and we have now addressed this in the manuscript. Our goal for this part of the study was to investigate whether ScaC mimics an endogenous cargo to bind BICD2. We show that introducing mutations that block the binding of Rab6 and RanBP2 does not affect ScaC binding, which suggests that ScaC does not mimic Rab6 or RanBP2 to interact with BICD2. Over the past two years a third BICD2 interactor, Nesprin-2, has been described, however its binding site on BICD2 remains unknown. Because of this, we are not able to conclude that Nesprin-2 and ScaC bind to different motifs on BICD2.

We have made multiple attempts to solve the structure of the BICD2-ScaC complex using X-ray crystallography. Unfortunately, these attempts were ultimately unsuccessful, and we were unable to fully identify the ScaC-binding site on BICD2.

3.9) What is the impact of knocking down BICD1 and BICD2 on microtubule localization and structure? Does *O. tsutsugamushi* and/or still localize with and/or activate microtubules without BICD2?

We examined the morphology of the microtubule cytoskeleton in HeLa Flp-In cells upon BICD1 and BICD2 knockdown by immunofluorescence staining of alpha-tubulin. As shown in the figure below, we observed no difference in microtubule structure or localization between the knockdown and control conditions. This indicates that the phenotype we observed in our mitochondrial relocation assay is not attributable to an indirect effect of BICD1/2 knockdown on microtubule stability.

Our finding (shown in Figure 4E-F) that double BICD1/BICD2 knockout prevents perinuclear trafficking of the bacterium, phenocopying the effects of nocodazole treatment, indicates that BICD adaptors are required for *O. tsutsugamushi* to undergo transport along microtubules.

Minor

3.10) Line 34-37, *Chlamydia trachomatis* does not move along microtubules. Rather, the chlamydial inclusion traffics on microtubules. Also, chlamydiae are not encased in endocytic membranes per se. It is better to refer to it as the inclusion membrane.

We thank the reviewer for this clarification. We have amended the text in lines 38-41 of the introduction to read: “Whereas *Chlamydia* remains encased within an inclusion membrane derived from the endocytic compartment and leverages the host vesicular transport system, *O. tsutsugamushi* resides free in the cytosol and therefore needs to establish direct connections with microtubule motors.”

3.11) What is the level of conservation between ScaC and other Sca proteins? This is important to consider if ScaC is the unique Sca to bind BICD1/2 and why other Sca proteins should be included as controls. Such information could help inform experiments to pin down the region of ScaC required for binding BICD1/2.

We thank the reviewer for raising this valid point. It is important to stress that while all four Sca proteins share a conserved C-terminal translocator domain, their passenger domains are highly diverse in terms of length, amino acid sequence and secondary structure. For instance, the passenger domain of ScaC has a predicted coiled-coil structure whereas that of ScaA mainly consists of beta-sheets. To illustrate this point, we have performed structure predictions of the Sca passenger domains using AlphaFold3 (below).

Based on the above, we do not expect functional redundancy between different Sca proteins. We nonetheless included the passenger domain of ScaE as a control in our *in vitro* binding assay, since it shares the closest resemblance to ScaC in terms of secondary structure. We purified the passenger domain of ScaE in *E. coli* and checked if it formed a complex with BICD2 using size exclusion chromatography. As we show in Supplementary Figure 3E, the two proteins eluted independently and did not interact *in vitro*.

3.12) The circular dichroism data should be better explained in the results and/or legend to be digestible by a broader audience.

We have expanded the legend for Figure S3A to provide additional details about the technique and the results it generates. It now reads: “*Circular dichroism analysis of purified ScaC. The plot shows ellipticity (mdeg) as a function of wavelength (nm), and the secondary structure of the protein is inferred from the shape and magnitude of the ellipticity curve. The annotated dips at 208 nm and 222 nm suggest that the passenger domain of ScaC is predominantly alpha helical.*”

3.13) Lines 186-188 are confusing. Because BICDR1 has a Rab6 binding site is the very reason it cannot interact with ScaC???

BICD2 and BICDR1 share a conserved Rab6-binding site in their C-terminus. If ScaC had evolved to bind BICD2 by mimicking Rab6, we would expect it to also bind BICDR1. The finding that ScaC does not bind BICDR1 supports our results, now in Figure 2, which show that ScaC does not interact with BICD2 in exactly the same way as either Rab6 or RanBP2. We have moved the BICDR1 observation before we introduce the Rab6 and RanBP2 mutants and reworked the sentences to reduce confusion (Lines 181-183).

Reviewer #4 (Remarks to the Author):

In this manuscript, Manigrasso et al. demonstrated that *O. tsutsugamushi* recruits the activating adaptors BICD1 and BICD2 for dynein-based retrograde trafficking. Through SEC and single molecule assays, they showed that the outer membrane passenger domain of ScaC from *O. tsutsugamushi* interacts with the CC3 domain of BICD2, releasing its auto-inhibition and making its N-terminus available for interaction with dynein and dynactin. They also demonstrated that BICD2 is necessary for the nuclear migration of *O. tsutsugamushi*.

The *in vitro* data are of high quality, and the system and findings hold significant potential for the field. However, there are a few minor caveats that are not adequately addressed, and some important controls related to the key experiments need to be conducted.

4.1) Since the ScaB autotransporter is essential for the invasion of mammalian cells by *Orientia tsutsugamushi*, the authors need to provide a more robust explanation for why they did not study the GFP-ScaB co-IP/MS in HeLa cells.

We focused our analysis to the four Sca proteins that are conserved in all currently sequenced *Orientia* strains. ScaB has been described from strain Boryong but is not

present in many *O. tsutsugamushi* strains that undergo transport to the perinucleus, such as Gilliam, Kato, UT144, TA763, UT76 and Ikeda. This suggests it is not required for this process. We now explain that we focused on those Sca proteins “ubiquitously conserved across *O. tsutsugamushi* strains which undergo transport to the perinucleus” (Lines 88-91).

4.2) Cell lines such as monocytes/macrophages and dendritic cells are more biologically relevant hosts for *O. tsutsugamushi*. To avoid overinterpreting the results, the overexpression of ScaA/C/D/E and its interaction with BICD1/2 should be reported in at least one of these cell lines.

We have performed pull-downs using purified ScaC protein and whole-cell lysates from monocytic (THP-1) and dendritic cell (MutuDC) lines and show that ScaC interacts with BICD1 and BICD2 adaptors in both cell types. These results are presented in Supplementary Figure S1B-C of the revised manuscript and referred to in line 98-101.

4.3) Furthermore, in supporting figure 4, the authors have shown that BICD1 has almost zero expression in HeLa cells (Fig S4 K) and they showed the gene expression level of BICD2, measured by qPCR, are ~57-fold higher than those of BICD1 in the cell model. Therefore, its knockdown in HeLa cells cannot provide conclusive results regarding ScaC trafficking. At least one more cell line with higher BICD1 expression needs to be included in this study (such as lymphatic endothelial cells that they mentioned in the discussion part).

According to the Human Protein Atlas, BICD1 expression is mainly confined to the brain. We were unable to identify a relevant cell line where BICD1 is more abundant than BICD2. We checked *BICD1* and *BICD2* expression in HUVEC cells, which are a model system for endothelial cells, but found that BICD2 is still more abundant than its paralog (see figure below).

To determine whether BICD1 can (at least in principle) mediate the transport of ScaC, we attempted to rescue the mitochondrial dispersion phenotype caused by BICD2 depletion in GFP-ScaC^{MTS} cells by overexpressing a mCherry-BICD1 cassette. We show that BICD1 can compensate for BICD2 in our mitochondrial relocation assay, indicating that the two adaptors are functionally equivalent (Supplementary Figure S2M-N).

Furthermore, as also described above, we performed individual and double knockouts of BICD1 and BICD2 in HeLa ATCC cells and found that bacterial movement to the perinucleus was completely disrupted only in the double knockout line (see Figure 4E-F of the revised manuscript). This result indicates that BICD1 actively participates in bacterial transport even in cell lines where its expression is minimal.

4.4) The results indicate that GFP-ScaCMTS expression mimicked the effect of GFP-BICD2NMTS. Therefore, the authors need to provide data showing the localization of over-expressed GFP-ScaC itself (without mitochondrial attachment) within the cell. Can they observe any perinuclear accumulation? Additionally, why do mitochondria bound to ScaC accumulate at 1-3 single perinuclear points? Shouldn't they be distributed around the nucleus?

(i) We have carried out IF analysis of HeLa Flp-in cells overexpressing GFP-BICD2N and GFP-ScaC without mitochondrial attachment. As shown in the figure below, GFP-BICD2N was predominantly diffused in the cytoplasm with the exception of a small perinuclear cluster colocalizing with the centrosome (stained with a pericentrin marker). GFP-ScaC mimicked this localization pattern and colocalized with endogenous BICD2.

Why a clear BICD2N accumulation at the perinucleus is only observable upon organelle attachment remains enigmatic. However, it is a known phenomenon in the field (see Fig. 9 of Hoogenraad *et al.* 2001, PMC149157 and Fig. 1 of Hoogenraad *et al.*, 2003, PMC275447), and is the reason why organelle relocation assays are routinely used to determine if a protein is a dynein effector.

(ii) Dynein moves its cargoes towards the minus ends of microtubules. In fibroblasts, microtubules are generally organized into a radial array where the minus ends converge at the centrosome (which is located near the nucleus), which functions as a microtubule organizing centre (MTOC). Because of this, we would expect dynein-driven cargoes (in this case ScaC-bound mitochondria) to accumulate at a single perinuclear point corresponding to the MTOC.

4.5) Does this substantial accumulation of GFP-ScaCMTS (mitochondria) at the nucleus impact the distribution of other organelles within the cells? The authors should provide evidence that cells remain healthy upon overexpression of ScaC.

- (i) We looked at the localization of ER, lysosomes and Golgi vesicles in GFP^{MTS} and GFP-ScaC^{MTS} cells by IF (using antibodies against Calnexin, LAMP1 and TGN46 respectively). Representative images are shown in the figure below. We noticed that in GFP-ScaC^{MTS} cells the organelles were excluded from the perinuclear region (indicated by white arrows) which is occupied by mitochondria. Apart from this, we did not observe any major changes in the localization pattern of these organelles. We further looked in GFP and GFP-ScaC cells (without mitochondrial attachment) and again we did not detect any significant changes in subcellular distribution of ER, lysosome and Golgi vesicles. We concluded that ScaC overexpression is not itself affecting the localization of these organelles.
- (ii) To check that overexpression of ScaC is not toxic to cells, we performed a growth curve of GFP-ScaC cells and showed that they divide at similar rates as control GFP cells (see Supplementary Figure S2D). Furthermore, we show that mitochondria are not clustered in GFP-ScaC cells (see Supplementary Figure S2E), indicating that ScaC overexpression is not indirectly causing the perinuclear accumulation of mitochondria that we observed in GFP-ScaC^{MTS} cells.

4.6) The impact of BicD2 knockdown on nuclear localization (as shown in Figure 2) appears to be relatively minor. The authors should also present the results of the BicD2 knockout experiment.

As suggested by the reviewer, we generated individual and double knockout lines for BICD1 and BICD2 in HeLa ATCC cells. BICD1 knockout had no noticeable effect on the perinuclear localization of *O. tsutsugamushi*, whereas BICD2 knockout caused only a mild defect in bacterial translocation. In contrast, the double BICD1/BICD2 knockout fully disrupted perinuclear transport of *O. tsutsugamushi*, leaving the bacteria scattered throughout the cytosol. As described above, these findings are presented in Figure 4 of the revised manuscript.

4.7) Furthermore, it is essential to report a “direct” infectivity phenotype following BicD2 knockdown/out, such as deficiencies in replication or the number of daughter bacteria formed inside the cell.

As mentioned above, we have now shown that blocking the trafficking of *O. tsutsugamushi* to the perinucleus via ScaC overexpression does not affect bacterial replication under the conditions we tested (see Supplementary Figure 6 C). The natural lifecycle of *Orientia tsutsugamushi* is substantially more complex than that tested in our experimental set up and we interpret the lack of phenotype to reflect a difference between our experimental conditions and the natural growth and transmission cycles of this bacterium.

Orientia's vector and reservoir are lepto-trombidium mites, and it is possible that blocking perinuclear trafficking would cause a growth defect in this organism (however, there are no mite cell lines available for us to test this hypothesis). Moreover, it is possible that dynein-based movement confers a growth advantage only under specific conditions.

A new manuscript from our lab, recently posted on Biorxiv, supports a hypothesis whereby loss of perinuclear trafficking may be disadvantageous during infection *in vivo* despite not causing a growth phenotype in cultured cells. In that study we compared the virulence of multiple strains of *O. tsutsugamushi* using a mouse model and human epidemiological data. We identified one strain, TA686, as being avirulent in humans, non-human primates, and mice. We sought to identify the basis of this and found that one of the major noticeable differences between TA686 and other strains was a lack of strong perinuclear localisation in cultured cells. We measured the relative growth rate of multiple strains in multiple cell lines and found no correlation between *in vitro* growth and *in vivo* virulence. Whilst the data does not prove that the reduced ability to traffic to the perinucleus in TA686 causes its lack of virulence, it does support our current interpretation in two ways: 1. growth rate *in vitro* does not reflect ability to disseminate and cause disease or be cleared by the host immune response *in vivo*, and 2. the one strain that is unable to cause disease is also defective in perinuclear trafficking compared to others.

We address all these points in the Discussion section of the revised manuscript. Additionally, we plan to carry out further studies to determine the role of perinuclear trafficking *in vivo* but this is far out of the scope of the current report.

4.8) Btw, authors need to clarify what TSA56 represents as a phenotype of bacterial infection? There is no information provided on this in the results section.

TSA56 is the major surface protein of *O. tsutsugamushi* that is highly abundant and encases the bacterium. We routinely use antibodies against TSA56 to label the bacterium for microscopy studies. We now say this in the results section.

4.9) Although cross-linking experiments demonstrated that residues 119-136 of ScaC interact with the central region of BicD2 CC3, the researchers opted to study the

interaction between truncated ScaC (residues 50-159) and BicD2 in more detail. The researchers need to report the cross-linking results for residues 119-136 of ScaC with BicD2-CC3.

The match scores for our cross-linking experiments are listed in Supplementary Table 5. We tried to narrow down the region of ScaC that interacts with BICD2, however we were unable to express and purify ScaC truncations shorter than our 50-159 construct.

4.10) In Figure 4A, it appears that the authors used the incorrect colors for the lines in the chromatogram. The red line should correspond to "BICD2N + BICD2C + ScaC," while the black line should represent "BICD2N + BICD2C" only.

We thank the reviewer for spotting this mistake, which we have now corrected.

4.11) "As purified viral capsids were found to interact with the BICD2 CC3" → Carnes et al (ref 47) weren't able to show the binding of CC3 to the assembled capsid. Line 267 needs to be corrected accordingly.

This was a mistake, only the reference Dharan et. al. showed the interaction of viral capsids with BICD2 CC3. We have corrected this.

REVIEWERS' COMMENTS

Reviewer #2 (Remarks to the Author):

The authors have revised the manuscript with additional experiments, which has sufficiently addressed my concerns to the initial version of the manuscript. The manuscript is well-written. The new experimental data with the BICD adaptors knock-out cells supported the role of dynein (via the interaction with BICD1 and BICD2) on perinuclear clustering of the bacteria.

The revised manuscript offers new insights on how bacterial pathogens exploit the host molecular motors for establishing their intracellular niche. The reported observations will stimulate further discovery of the field.

Thank you for your comments.

Reviewer #3 (Remarks to the Author):

I appreciate the authors' response to my and the other reviewers' concerns. The additional data, particularly the use of BICD1 and BICD2 KO lines, nocodazole, and dominant negative ScaC in the context of infection when combined with the strong biochemical data make for a compelling story that the ScaC-BICD1/2 interaction contributes to *Orientia* mobilization along microtubules. I have only minor comments at this point.

Thank you for your comments.

1. I am still unconvinced by the response that freezing host cell-free *Orientia* does not impact its viability/fitness. First, the Giengkam 2015 paper does not directly compare the growth of *Orientia* in cells after being frozen in various buffers to *Orientia* that had not been frozen (Fig 7C). Second, in looking at the IF images in the response to critique 2.1, at 2 dpi there are barely any bacteria in the cell. In fact, I would argue that there are 1-2 log-fold differences in the number of bacteria in these cells vs if the inocula had come from actively growing *Orientia* infected cells. If your MOI was 50 and you only see 4-5 intracellular bacteria at 2 dpi, then the viability of your inoculum was compromised.

That said, the *Orientia* load in WT cells in Figure 4E is much higher than in the aforementioned data. Because 4E data is what was included in the manuscript and it supports the hypothetical model, I won't belabor this point further. At some point, however, I suggest that you directly compare the growth of equivalent numbers of live *Orientia* recovered from infected cells vs frozen host cell-free *Orientia*. Adding so much dead frozen and thawed bacteria is a concern and could ultimately yield misleading results depending on what you are examining, especially if you ever look at host immune responses.

The Salje lab routinely carry out experiments on both fresh and frozen bacterial stocks and get similar results, so we do not think this is likely to affect the results or conclusions in this work.

2. Lines 204-206 (Fig. 3A) – After reading and looking over Figure 3A several times, I do not see how the chromatogram supports this statement. Please revise to provide more detail or clue me in.

We have rewritten the description of the experiment shown in Figure 3A and adjusted the figure legend to make this clearer. In summary: to detect the interaction between the N and C termini of BICD2 by SEC it is necessary to express them as separate constructs. Mixed together they form a complex as shown by the fact they elute together at a larger size (smaller elution volume) than the two components individually. The top gel shows that the maximum intensity of the bands for BICD2C and BICD2N are in the same fraction also demonstrating co-elution. Upon addition of ScaC the BICD2N dissociates from BICD2C as shown by the fact that the maximum intensity of its band shifts to a lower molecular weight fraction (larger elution volume). Under these conditions ScaC coelutes in the same fraction as BICD2C showing the two interact.

3. Line 52 – I'm not sure if phagosomal is the best adjective here given *Orientia* endocytic entry. Perhaps endosome. Or if you want to go more generic to toe the line between endosome and phagosome, then perhaps vacuolar.

We have changed phagosomal to endosomal.

4. Line 129 – I suggest moving (Fig. 1H-I) to the end of the sentence on line 132.

Done

5. Supplementary figure 2L – italicize GAPDH since you are measuring gene expression.

Done

6. Lines 213, 218, 230 – you pick and choose when you want to list values. I'm not sure this is necessary especially for the 3-fold difference (lines 213-124, Figure 3B-C) as the fold difference is clear via the bar graph. Ultimately, this is up to you, and is a suggestion.

We have kept the values as they are helpful for those who study microtubule motors.

7. 249-250 – If this manuscript moves forward, the Nat Comm copy editors will likely weigh in on this, but I do not think you can refer to data and not show it. I suggest adding to Supplementary figure 6 as it does easily set up why you went for the CRISPR knock outs of BICD1/2.

We have added the si-RNA knockdowns back into Supplementary Figure 6 as suggested.

8. Lines 280-283 – Similarly, I do not think you can reference a BioRXIV manuscript as it has not been peer reviewed. I suggest striking this information and reference unless it gets published in time.

The Editor said that citation of bioRxiv manuscripts is allowed.

Reviewer #4 (Remarks to the Author):

I appreciate the authors for addressing all my concerns. They have conducted the related experiments excellently, and I strongly recommend the current version of this manuscript for publication in Nature Communications.

Thank you for your comments.